



# Ecological and environmental controls on plant wax production and stable isotope fractionation in modern terrestrial Arctic vegetation

Kurt R. Lindberg[1], Elizabeth K. Thomas[1], Martha K. Raynolds[2], Helga Bültmann[3], and Jonathan H. Raberg[4]

[1]Department of Earth Sciences, University at Buffalo, Buffalo, NY, 14260, USA
[2]Institute of Arctic Biology, University of Alaska Fairbanks, Fairbanks, Alaska, 99775, USA
[3]Institute of Landscape Ecology, University of Münster, Münster, D-48149, Germany
[4]Department of Geology and Geophysics, University of Wyoming, Laramie, Wyoming, 82072, USA

**Correspondence:** Kurt R. Lindberg (kurtrlindberg@gmail.com)

**Abstract.** Terrestrially-derived plant waxes and their compound-specific stable carbon ($\delta^{13}$C) and hydrogen ($\delta^{2}$H) isotope ratios are valuable tools for inferring past changes in vegetation and hydrology in sedimentary archives. Such inferences require knowing the ecological (i.e. plant growth form) and environmental (i.e. temperature, precipitation, relative humidity, elevation) mechanisms that govern the production of different plant wax carbon chain-lengths and the fractionation of stable isotopes. These mechanisms, however, are uncertain in the Arctic, limiting our ability to infer past vegetation and hydrology changes. To address this, we produced terrestrial plant $n$-alkanoic acid and $n$-alkane abundance and $\delta^{13}$C and $\delta^{2}$H data from a latitudinal environmental gradient along the Eastern Canadian Arctic (105 individuals), which we compiled with published data across the Arctic (additional 281 individuals). We compared this dataset with environmental parameters to assess the mechanisms that govern plant-wax production and isotope fractionation. We found that total plant wax concentrations and Average Chain-Length (ACL) were statistically different between vascular (trees, shrubs, forbs, ferns, graminoids) and non-vascular plants (mosses, liverworts) and lichens, whereas $\delta^{13}$C values and $\delta^{2}$H apparent fractionation relative to growing season precipitation $\delta^{2}$H often did not differ significantly between plant growth forms. Correlations between plant wax indices and mean of the months above freezing (MAF) environmental parameters were generally weak ($r \leq 0.4$), and/or not significant ($p > 0.05$). These results suggest that a fundamental assumption to paleoclimate research holds in the Arctic: for individual plant taxa and plant communities, the abundance, ACL, and $\delta^{13}$C/$\delta^{2}$H isotopic fractionation of both $n$-alkanoic acids and $n$-alkanes is independent of temperature, precipitation, humidity, and elevation. Instead, changes in sedimentary plant wax distributions reflect changes in plant taxa present through time, and changes in plant wax $\delta^{2}$H reflect changes in source water $\delta^{2}$H. Therefore, plant waxes can be used to infer past changes in climate and ecology.

## 1 Introduction

Anthropogenic activities are driving rapid climate change and associated environmental changes in the Arctic, which are projected to continue through the end of this century (IPCC, 2023). These changes include increasing temperatures, increasing precipitation amount, and altered precipitation seasonality, as well as the northward expansion of lower-latitude vegetation into





Arctic tundra biomes (Bintanja and Selten, 2014; Box et al., 2019; Elmendorf et al., 2012). Plant waxes and their compound-specific stable isotopes are valuable tracers of past ecological and hydrological change in sedimentary archives (Liu et al., 2022;

Sachse et al., 2012), providing crucial analogues for environmental changes observed today. Interpretations of changes in these proxies over time are often based on our understanding of plant waxes produced by modern vegetation across a wide range of plant types and growing conditions. However, these modern datasets and our understanding of factors like chemotaxonomy and environmental conditions that influence plant wax proxies are primarily derived from temperate and tropical vegetation (Bush and McInerney, 2013; Diefendorf and Freimuth, 2017; Gao et al., 2014; Sachse et al., 2006). Expanding these modern

datasets into Arctic biomes will improve our ability to reconstruct past Arctic change using sedimentary plant waxes.

Plant waxes, including $n$-alkanoic acid, $n$-alkane, and $n$-alcohol compounds, are straight-chain hydrocarbons (carbon chain-length $\geq 20$) produced on the surfaces of plants to regulate moisture balance and ultraviolet light absorption (Eglinton and Hamilton, 1967; Post-Beittenmiller, 1996; Yeats and Rose, 2013). The distribution of plant wax carbon chain-lengths in modern terrestrial vegetation may be specific to different plant types (Bush and McInerney, 2013; Ficken et al., 2000; Liu et al.,

2022). For example, in a synthesis of 87 geographically-diverse published datasets, Bush and McInerney (2013) demonstrated that *Sphagnum* mosses generally produce a greater portion of mid-chain $C_{23}$ and $C_{25}$ $n$-alkanes whereas trees, woody plants and grasses produce more long-chain $C_{27}$, $C_{29}$, and $C_{31}$ $n$-alkanes. Liu et al. (2022) also found significant differences in plant wax chain-length distribution indices, such as Average Chain-Length (ACL) and $C_{29}/(C_{29} + C_{31})$ between globally-distributed woody and non-woody terrestrial vascular plants. However, this relationship may not be as robust in Arctic vegetation com-

munities. In the Arctic, shrubs (e.g., *Betula* sp.) and graminoids (e.g., *Carex* sp.) produce a substantial portion of mid-chain $n$-alkanes and $n$-alkanoic acids (Berke et al., 2019; Dion-Kirschner et al., 2020; Hollister et al., 2022). These findings complicate the interpretation of Arctic plant wax chain-length distributions.

The ratios of modern plant wax compound-specific stable carbon ($\delta^{13}$C) and hydrogen ($\delta^2$H) isotopes also vary between plant growth forms. Terrestrial plant wax carbon is sourced from atmospheric $CO_2$, with biosynthesis fractionating against the

heavier isotope, $^{13}$C. The carbon isotope fractionation difference between $C_3$ (more $^{13}$C-depleted) and $C_4$ (less $^{13}$C-depleted) photosynthetic pathways has been well documented, and is commonly used to reconstruct past vegetation change between these two broad plant community types (Cerling and Harris, 1999), although $C_4$ plants do not occur at high latitudes. Diefendorf and Freimuth (2017) also noted variations in plant wax $\delta^{13}$C between growth forms within the same photosynthetic pathway: $C_3$ trees are $^{13}$C-enriched compared to $C_3$ shrubs and forbs.

Similarly, terrestrial plant wax hydrogen is sourced from meteoric water stored in the soil. This source water experiences fractionation during evaporation in soil and within the plant, and subsequently during biosynthesis (Sachse et al., 2012). The net apparent fractionation ($\epsilon_{app}$) between source water and plant wax $\delta^2$H values incorporates all of these fractionation processes as well as geographical variation in precipitation $\delta^2$H values. $\epsilon_{app}$ also varies between plant growth forms sampled from the same locale, likely due to physiological differences in water use efficiency and metabolic pathways (Gao et al., 2014; Kahmen et al.,

2013; Saishree et al., 2023). Understanding these fractionation differences between plant types in both carbon and hydrogen stable isotope systems is critical for disentangling past vegetation change from other reconstructed climatic parameters.



Environmental factors, including temperature, precipitation amount, and relative humidity, may also affect how individual plant taxa produce different plant wax chain-lengths and fractionate stable carbon and hydrogen isotopes. For example, positive relationships exist between ACL and mean annual temperature within certain *Acer* sp. and *Juniperus* sp. tree species along the

eastern United States (Tipple and Pagani, 2013). Similarly, the ACL of *Acacia* sp. and *Eucalyptus* sp. are significantly correlated to annual precipitation amount and relative humidity in northern Australia; though the former relationship was positive and the latter negative (Hoffmann et al., 2013). Plant wax $\delta^{13}$C has a negative relationship with precipitation amount across a global range of biomes (Diefendorf et al., 2010). Additionally, $\epsilon_{app}$ in Chinese monocot and dicot plants also has negative relationships with annual precipitation amount (Liu et al., 2023).

Variation in plant wax data, both between plant growth forms and within individual taxa across environmental gradients, creates more uncertainty in paleoclimate reconstructions when both taxonomic and environmental factors are not well constrained for a particular study area. For example, based on the findings described above, a change in ACL over time could represent either a change in the local plant community or a stable plant community responding to environmental change. Describing whether these plant wax indices respond to plant community and/or to environmental change along modern environmental

gradients in the Arctic will provide clarity when interpreting sedimentary plant wax data.

The Eastern Canadian Arctic (ECA), including Baffin Island, Nunavut and Nunavik, northern Quebec, contains a strong latitudinal climate gradient within which environmental controls on plant wax production and stable isotope fractionation may be assessed. This gradient is heavily influenced by oceanic currents (Briner et al., 2006). The West Greenland Current delivers warm, saline, subarctic water from the lower latitudes in the North Atlantic up the southwest coast of Greenland and curls

westward across Davis Strait and back down along southeastern Baffin Island towards Newfoundland and Labrador, creating a warmer, sub-Arctic climate in that region (Münchow et al., 2015). The Baffin Island Current brings cold, polar water from the Arctic Ocean through the Nares Strait and the Canadian Archipelago, then along the northeastern coast of Baffin Island, resulting in an arid, polar climate north of Davis Strait (Rudels, 2011). Modern sea ice extends out to the Labrador Sea during the early spring maximum and retreats north of Baffin Island past the Nares Strait during the late summer minimum (Akers

et al., 2020). These conditions produce strong latitudinal temperature and precipitation gradients within the ECA, which are reflected in distinct bioclimate subzones (Fig. 1b) where sub-Arctic conifers and low Arctic shrub communities are limited to the south, and cryptogam-rich high Arctic tundra becomes the dominant vegetation community in the north (Walker et al., 2005).

In this study, we analyzed the chain-length distributions of plant wax *n*-alkanoic acids and *n*-alkanes along with the $\delta^{13}$C

and $\delta^2$H of *n*-alkanoic acids from terrestrial vascular (trees, shrubs, forbs, ferns, graminoids) and non-vascular plants (mosses, liverworts) and lichens collected across the ECA. We combine this new dataset with a compilation of published modern Arctic plant wax datasets, which span substantial gradients in temperature, precipitation amount, relative humidity, and elevation while focusing the analysis to areas including and north of the boreal forest. We examined whether variations in Arctic terrestrial plant wax data were primarily driven by individual plant growth forms responding to their environmental conditions, or by

differences in plant wax synthesis between growth forms. Evaluating these two potential drivers of modern Arctic plant wax variability is essential for assessing whether sedimentary plant waxes record or respond to past environmental change. In the





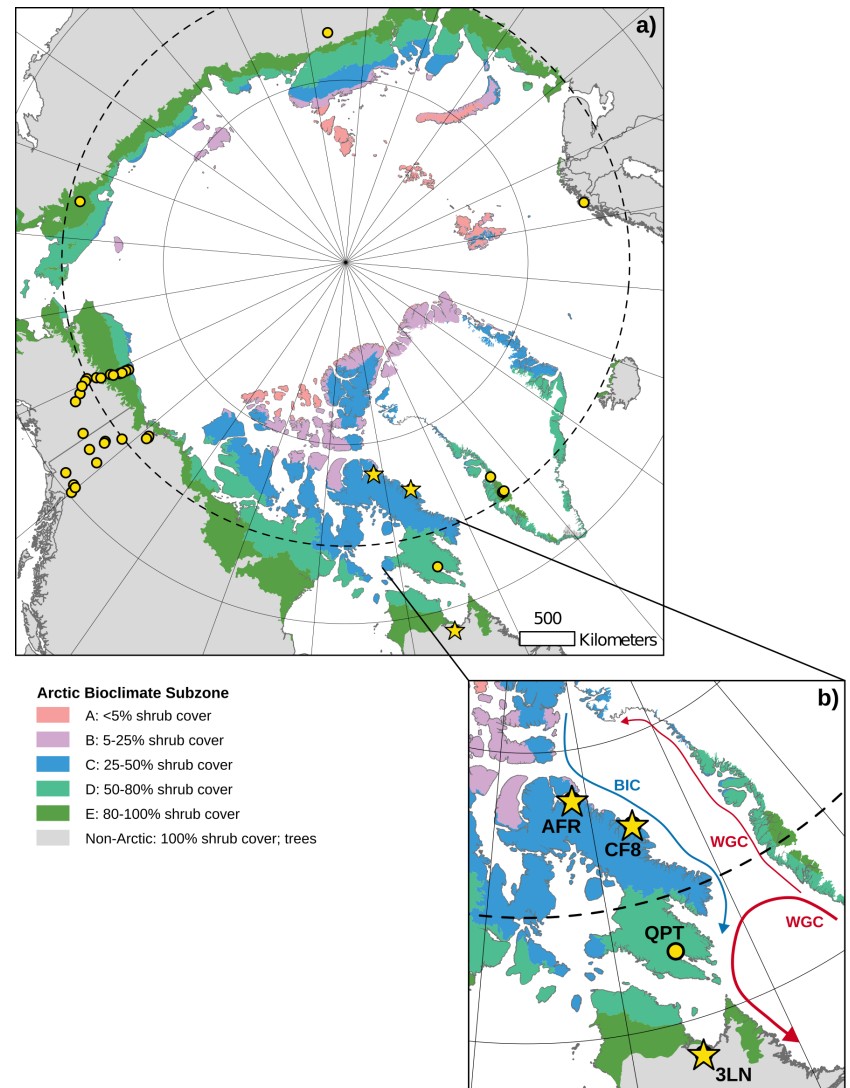

**Figure 1.** (a) Map of the circumpolar Arctic showing terrestrial plant sampling location for this study (yellow stars) and compiled, published datasets (yellow dots). (b) Enhanced view of Eastern Canadian Arctic (ECA) sampling locations. The blue arrow shows the flow of relatively cold water in the Baffin Island Current (BIC) and the red arrows show the flow of relatively warm water in the West Greenland Current (WGC). Shaded regions in both panels represent different Arctic bioclimate subzones (A-E; Non-Arctic) as defined by the Circumpolar Arctic Vegetation Map (Walker et al., 2005).

context of this study, "recording" refers to plants not changing their plant wax production, or stable isotope fractionation (i.e. an individual always produces the same chain-length distributions and fractionates source water hydrogen isotopes to the same degree). Plants "responding" to their environment means the opposite, where different climate settings alter an individual's

plant wax properties.





## 2 Materials and Methods

### 2.1 Study Area: Eastern Canadian Arctic and High-Latitude Data Compilation

We collected 105 terrestrial vegetation samples for plant wax analysis across three lake catchments spanning 14 degrees of latitude in the ECA during summer field seasons in 2019 and 2021 (Fig. 1b). Lake Africa (AFR; Informal name; 72.42 °N, 77.44 °°W; 895 masl) is located in northern Baffin Island and represents our northernmost sampling location in this study. Due to its high elevation, AFR resides in bioclimate subzone A and its catchment is dominated by mats of liverworts, mosses, and lichen and contains only one vascular plant species, a grass, *Phippsia algida*. Lake CF8 (Informal name; 70.56 °N, 68.95 °W; 195 masl) is located in the Clyde Foreland of northeastern Baffin Island and is characterized as bioclimate subzone C. While lichens and cryptogam mats are still the most common vegetation in the catchment, prostrate dwarf shrubs (e.g. *Cassiope tetragona*) and graminoids (e.g. *Luzula confusa*) are also present (Thomas et al., 2023). Lake 3LN (Informal name; 58.10 °N, 68.46 °W; 61 masl) is on the Canadian mainland in the Nunavik region of northern Quebec. In contrast to the previous two lakes, Lake 3LN is within the tree line and is classified as a sub-Arctic bioclimate. The catchment contains a wide variety of vascular plant vegetation, including trees (*Larix laricinia*, *Picea mariana*), shrubs (e.g. *Betula glandulosa*, *Rhododendron* sp., *Salix* sp., *Alnus viridis*), and graminoids (e.g. *Carex* sp., *Eriophorum* sp.). Bryophytes are also present in wetter areas of the catchment and are dominated by *Pleurozium schreberi* and *Sphagnum* sp. We also include previously published terrestrial plant wax data from Lake Qaupat (QPT; 63.68 °N, 68.20 °W; 33 masl) in southern Baffin Island for this ECA transect (Hollister et al., 2022). Positioned in bioclimate subzone D, the Lake QPT catchment is dominated by *Betula glandulosa* and *Salix* sp. shrubs, similar to Lake 3LN, but does not contain any trees. The ECA transects included 139 plant samples across seven major plant growth forms; trees ($n = 6$), shrubs ($n = 43$), forbs ($n = 1$), graminoids ($n = 17$), mosses ($n = 36$), liverworts ($n = 18$), and lichens ($n = 18$).

To expand the sample size and range of environmental conditions, we compiled published terrestrial plant wax data from sampling sites across the entire Arctic; within the latitude range spanned by the ECA transect (Fig. 1a). Regions added in this compilation include: west Greenland (Berke et al., 2019; Dion-Kirschner et al., 2020; Thomas et al., 2016), northern Norway (Balascio et al., 2018), northern Russia (Wilkie et al., 2013; Zibulski et al., 2017), and Alaska/Yukon/Northwest Territories (Bakkelund et al., 2018; Daniels et al., 2017; O'Connor et al., 2020). This compilation increased the number of unique sampling/environmental locations to 36, increased the total plant wax sample count to 386, and added ferns to the list of plant growth forms in this study; trees ($n = 18$), shrubs ($n = 183$), forbs ($n = 17$), ferns ($n = 15$), graminoids ($n = 52$), mosses ($n = 63$), liverworts ($n = 18$), lichens ($n = 20$). Different studies report different types of plant wax data, and most studies only report one plant wax compound class and/or one stable isotope, which limits the sample size for some analyses of individual plant wax data types. Compiled stable isotope data had to be accompanied by the concentration or relative abundance of each chain-length in order to calculate weighted average values for each sample (Section 2.3).





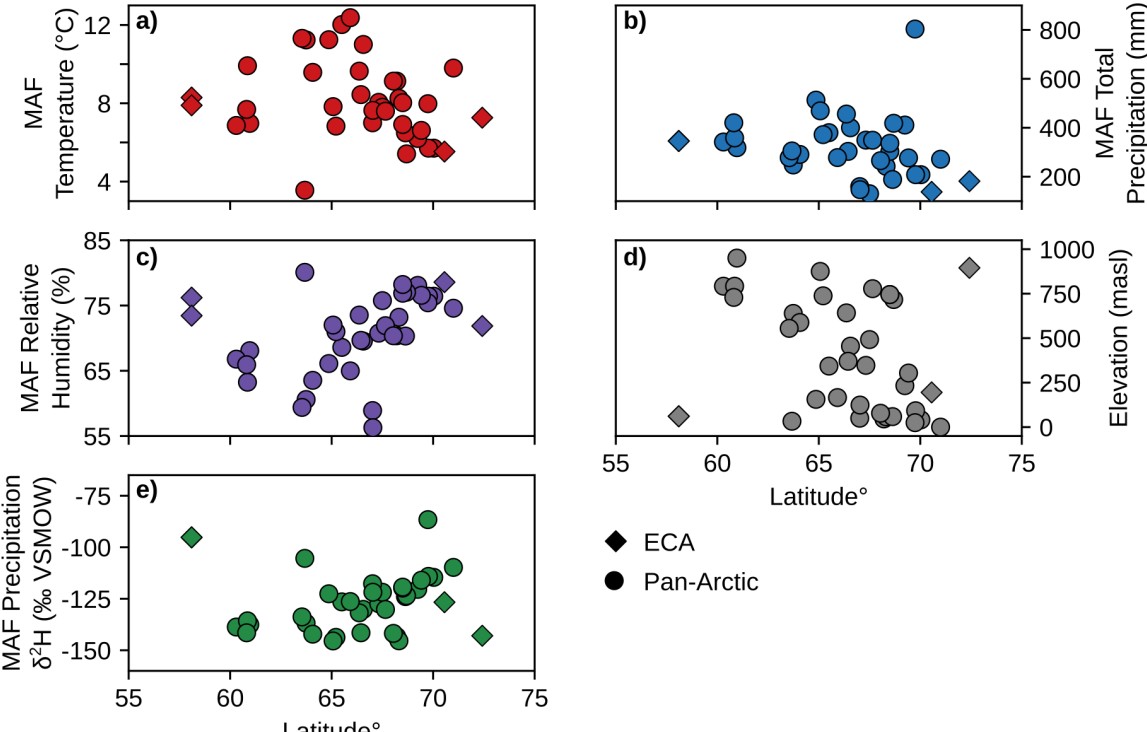

**Figure 2.** Scatterplots of environmental and isotopic parameters of the Months Above Freezing (MAF) for each unique pan-Arctic sampling location (Fig. 1) and sampling year. (a) Mean temperature. (b) Total precipitation amount. (c) Mean relative humidity % derived from dew point and temperature (Eq. 1). (d) Site elevation (meters above sea level; masl) (e) Precipitation amount-weighted mean precipitation isotope $\delta^2$H values. ECA sampling sites from this study (AFR, CF8, 3LN) are plotted as diamonds and all other pan-Arctic sites are plotted as circles. Temperature, precipitation, and relative humidity data in panels a-c are sourced from ERA5 reanalysis (Hersbach et al., 2020). Precipitation $\delta^2$H in panel d is sourced from the Online Isotopes in Precipitation Calculator (Bowen and Revenaugh, 2003; Bowen et al., 2005).

## 2.2 Environmental Parameters and Precipitation Isotopes

We used ERA5 reanalysis (Hersbach et al., 2020) as our data source for climate parameters at each sampling location and sampling year, including monthly temperature (Fig. 2a), precipitation amount (Fig. 2b), and relative humidity (Fig. 2c). Plant
waxes are produced yearly by individual plants, primarily during peak leaf flush in the early part of the growing season (Tipple et al., 2013), and therefore closely reflect the environmental conditions of the year they were sampled. The Arctic growing season, during which plant waxes are produced, is limited by the duration of temperatures above freezing (Bakkelund et al., 2018). We express the growing season environmental conditions of each plant sample as the mean of the months above freezing (MAF) for individual sampling years. Because ERA5 data can be extracted for the grid cell containing each study site, they are
preferable to station data, which in the ECA study area are few and all coastal, leading to discrepancies due to differences in distance, elevation, and other features (Gorbey et al., 2022). Since relative humidity ($RH$) is not directly provided by ERA5,



we calculated it for each month using ERA5 temperature ($T$) and dew point temperature ($D_p$) in Eq. 1 (Alduchov and Eskridge, 1996). We also compared plant wax indices to sample site elevation (Bakkelund et al., 2018; O'Connor et al., 2020), which was included in all compiled publications (Fig. 2d).

$$RH = 100 \times \left\{ e^{[17.625 \times D_p/(243.04+D_p)]} / e^{[17.625 \times T/(243.04+T)]} \right\} \tag{1}$$


### 2.3 Plant Wax Extraction, Quantification, and Stable Isotope Analysis

We extracted and quantified plant wax $n$-alkanoic acids and $n$-alkanes from 105 ECA plant samples using the standard methods of the University at Buffalo Organic and Stable Isotope Biogeochemistry Laboratory (Hollister et al., 2022). We measured $n$-alkanoic acid ($C_{20}$ to $C_{32}$) and $n$-alkane ($C_{21}$ to $C_{33}$) chain-length concentrations ($\mu$g/g dried plant) using a Trace 1310 Gas
Chromatograph-Flame Ionized Detector (GC-FID), with dual AI 1310 autosamplers and 30-m HP-1MS fused silica columns. We derived chain-length concentrations from GC-FID peak areas using external calibration curves of $C_{28}$ $n$-alkanoic acids and $C_{29}$ $n$-alkanes. We also measured for sample recovery during the full extraction and instrumental analysis processes using internal monounsaturated *cis-eicosenoic* $n$-alkanoic acid ($C_{20,1}$) and $C_{36}$ $n$-alkane standards. We examined the overall distributions of plant wax chain-lengths by calculating the Average Chain-Length (ACL; Eq. 2, 3) for each sample (Bray and
Evans, 1961; Bush and McInerney, 2013). We calculated the Carbon Preference Index (CPI; Eq. 4, 5) to describe the difference in production of even vs. odd and odd vs. even chain-lengths in $n$-alkanoic acids and $n$-alkanes, respectively (Marzi et al., 1993). CPI was not calculated for samples where the total abundance of chain-lengths in the denominator was equal to zero, either due to compounds being too small to quantify or not being reported in other publications (i.e. no odd-chain $n$-alkanoic acid data available).

$$ACL_{n-acid} = \Sigma(C_{n_{even}} \times n_{even}) / \Sigma(C_{n_{even}}) \tag{2}$$

$$ACL_{n-alk} = \Sigma(C_{n_{odd}} \times n_{odd}) / \Sigma(C_{n_{odd}}) \tag{3}$$

$$CPI_{n-acid} = [\Sigma(C_{20-30})_{even} + \Sigma(C_{22-32})_{even}] / [2 \times \Sigma(C_{21-31})_{odd}] \tag{4}$$

$$CPI_{n-alk} = [\Sigma(C_{21-31})_{odd} + \Sigma(C_{23-33})_{odd}] / [2 \times \Sigma(C_{22-32})_{even}] \tag{5}$$

We measured the compound-specific stable carbon ($\delta^{13}C$) and hydrogen ($\delta^2H$) isotope ratios of $C_{22}$ through $C_{28}$ even-chain
$n$-alkanoic acid chain-lengths in a subset of the ECA plant samples using the methods described in Hollister et al. (2022). Stable



isotope analysis was conducted in the University at Buffalo Organic and Stable Isotope Biogeochemistry Laboratory using a Thermo Delta V+ Isotope Ratio Mass Spectrometer (IRMS) with a split/splitless injector and a TriPlus RSH autosampler, connected to the IRMS via IsoLink II and Conflo IV. For $\delta^2$H analysis, we measured the $H_3^+$ factor at the beginning of each IRMS sequence, which ranged from $3.509 \pm 0.002$ to $5.005 \pm 0.038$ (mean $\pm 1\sigma$). We report $\delta^{13}$C and $\delta^2$H uncertainty
as the Standard Error of the Mean (SEM), accounting for uncertainty from instrument analysis and triplicate measurements of each sample. SEM uncertainty ranged from 0.1‰ to 0.6‰ with an average of $0.2 \pm 0.1$‰ for $\delta^{13}$C and 3.1‰ to 9.4‰ with an average of $4.2 \pm 1.3$‰ for $\delta^2$H. We calculated the net apparent $\delta^2$H fractionation ($\epsilon_{app}$) between amount-weighted precipitation $\delta^2$H values for the months above freezing and plant wax chain-length $\delta^2$H (Eq. 6). In Section 3, we report plant wax $\delta^{13}$C, $\delta^2$H, and $\epsilon_{app}$ as the abundance-weighted averages of $C_{22}$ through $C_{28}$ for $n$-alkanoic acids and $C_{23}$ through $C_{29}$
for $n$-alkanes. Doing so allows us to robustly compare stable isotope values between samples by accounting for the variability in isotope values and concentrations of individual chain-lengths. Average values hereafter are expressed as the mean $\pm 1\sigma$ standard deviation.

$$\epsilon_{app} = [((1000 + \delta^2 H_{plant\ wax})/(1000 + \delta^2 H_{source\ water})) - 1] \times 1000 \qquad (6)$$

### 2.4   Statistical Analyses of Plant Wax Data

We used Shapiro-Wilk tests to determine whether the values of plant wax indices within individual plant growth forms were normally distributed (Shapiro-Wilk test p-value $\geq 0.05$). We then employed Mann-Whitney U tests to evaluate whether chain-length or stable isotope values of different plant groups were significantly different (Mann-Whitney U test p-value $< 0.05$). We also used Pearson correlations to evaluate linear relationships between plant wax chain-length and stable isotope indices vs. scalar environmental parameters (temperature, relative humidity, precipitation amount, elevation). To examine the potential
influence of individual plant growth forms in this analysis, we perform multiple Pearson correlations for the same plant wax data type and set of environmental parameters while removing one growth form each time. We compared the resulting Pearson $r$ values to determine which growth form had the greatest impact on the correlation strength of the entire dataset when removed. For all statistical analyses, we used the averages of replicate species from the same location and sampling year (i.e. three *Alnus viridis* from Lake 3LN in 2021) to avoid overrepresentation of those samples (Bakkelund et al., 2018; Hollister et al.,
2022). We required each correlation to have a minimum of three observations per pair of variables. Shapiro-Wilk tests, Mann-Whitney U tests, and Pearson correlations were performed using the associated functions in the SciPy Python package v1.15.2 (Virtanen et al., 2020). These three statistical analyses were only performed on the pan-Arctic dataset due to the larger number of measurements per plant growth form and number of sampling sites for environmental comparisons.

We used Principal Component Analysis (PCA) to determine the primary modes of variability across all plant wax chain-
length distributions (i.e., even-chain $C_{20}$ to $C_{32}$ $n$-alkanoic acids and odd-chain $C_{21}$ to $C_{33}$ $n$-alkanes). Plant wax chain-length data naturally has a correlation bias because the relative abundance of chain length is constrained to a constant sum (Gloor et al., 2017). To address this, we apply a centered log-ratio (clr) transformation (Aitchison, 1982) on the plant wax chain-length data prior to PCA. Chain-length concentrations of 0 $\mu$g/g dry plant, either because chromatogram peaks were below the limit





of detection or were not reported in other publications, were not compatible when calculating the sample's geometric mean
for clr (Martín-Fernández et al., 2003). We replaced concentration values of zero with a small number equal to 1/N2 where
N is the number of plant wax chain-lengths used for PCA (Martín-Fernández et al., 2003). PCA was conducted using the
decomposition.PCA() function from the Scikit-learn project (Pedregosa et al., 2011). We performed PCA on plant wax data
from all ECA lakes (AFR, CF8, QPT, 3LN) and the pan-Arctic dataset.

## 3    Results

### 3.1    Plant Wax Chain-Length Abundances

We quantified plant wax $n$-alkanoic acid chain-length concentrations in 103 plant samples and $n$-alkane chain-length concentrations in 101 samples from Lake AFR, CF8, and 3LN in the ECA. Two $n$-alkanoic acid samples and four $n$-alkane samples
had concentrations that were below GC-FID detection. ECA plant wax chain-length concentrations, per gram of dried plant,
were highly variable. Total $n$-alkanoic acid concentrations ranged from 1.3 to 13,839 $\mu$g/g with an average of 745 $\pm$ 1,618 $\mu$g/g
(Fig. 3a), and $n$-alkane concentrations ranged from 0.5 to 6553 $\mu$g/g with an average of 625 $\pm$ 1,100 $\mu$g/g (Fig. 3b). Shrubs and
mosses produced the greatest total concentrations of the two plant wax compounds among vascular and non-vascular growth
forms, respectively. Liverworts produced the lowest total plant wax concentrations out of all growth forms. These patterns held
true for the pan-Arctic data compilation, though forbs (only $n$ = 1 for the ECA dataset) produced very high concentrations
of $n$-alkanes. Average pan-Arctic total $n$-alkanoic acid and $n$-alkane concentrations were 549 $\pm$ 1139 $\mu$g/g and 854 $\pm$ 2072
$\mu$g/g, respectively. Some plant growth forms had substantial differences between the total concentrations of each plant wax
compound. For example, ECA trees produced an average of 1297 $\pm$ 934 $\mu$g/g $n$-alkanoic acids but only 47.9 $\pm$ 18.5 $\mu$g/g $n$-
alkanes, while lichens produced 46.6 $\pm$ 54.4 $\mu$g/g $n$-alkanoic acids but 546 $\pm$ 888 $\mu$g/g $n$-alkanes. Shapiro-Wilk tests showed
that all but one plant growth form in each compound class, shrub $n$-alkanoic acids and fern $n$-alkanes, were log-normally distributed (Table S1). Mann-Whitney U tests showed that total plant wax $n$-alkanoic acid and $n$-alkane concentrations of shrubs
and lichens were significantly different from most other plant growth forms (Fig. 4a, b).

We calculated ACL and CPI using the full range of plant wax chain-lengths measured; $C_{20}$-$C_{32}$ for $n$-alkanoic acids, $C_{21}$-
$C_{33}$ for $n$-alkanes. ACL in ECA terrestrial plant wax $n$-alkanoic acid samples ranged from 21.5 to 29.3 with an average of 24.2
$\pm$ 1.5 (Fig. 3c), and $n$-alkane ACL ranged from 21.4 to 31.0 with an average of 26.9 $\pm$ 1.9 (Fig. 3d). Trees and shrubs produced
the greatest proportion of longer chain-lengths, overall, for both compound classes while liverworts and lichens produced the
most mid-chain waxes. However, some trees and shrubs, such as *Picea mariana*, *Cassiope tetragona*, and *Betula glandulosa*,
did produce high relative abundances of mid-chain waxes. Average pan-Arctic $n$-alkanoic acid and $n$-alkane ACL were 24.9
$\pm$ 1.7 and 27.2 $\pm$ 1.9, respectively. Shrubs were the only plant growth form with a non-normal distribution of $n$-alkanoic acid
ACL values, whereas only half of the plant growth forms had normally-distributed $n$-alkane ACL values (Fig. 3c, d). Liverwort
and lichen ACL values were the most significantly different among plant growth forms (Fig. 4c, d). $n$-alkanoic acid ACL
distributions in all non-vascular growth forms were significantly different from trees, shrubs, and forbs (Fig. 4c). CPI in new
ECA terrestrial plants was highly variable, ranging from 1.5 to 135 with an average of 12.7 $\pm$ 20.4 for $n$-alkanoic acids (Fig.





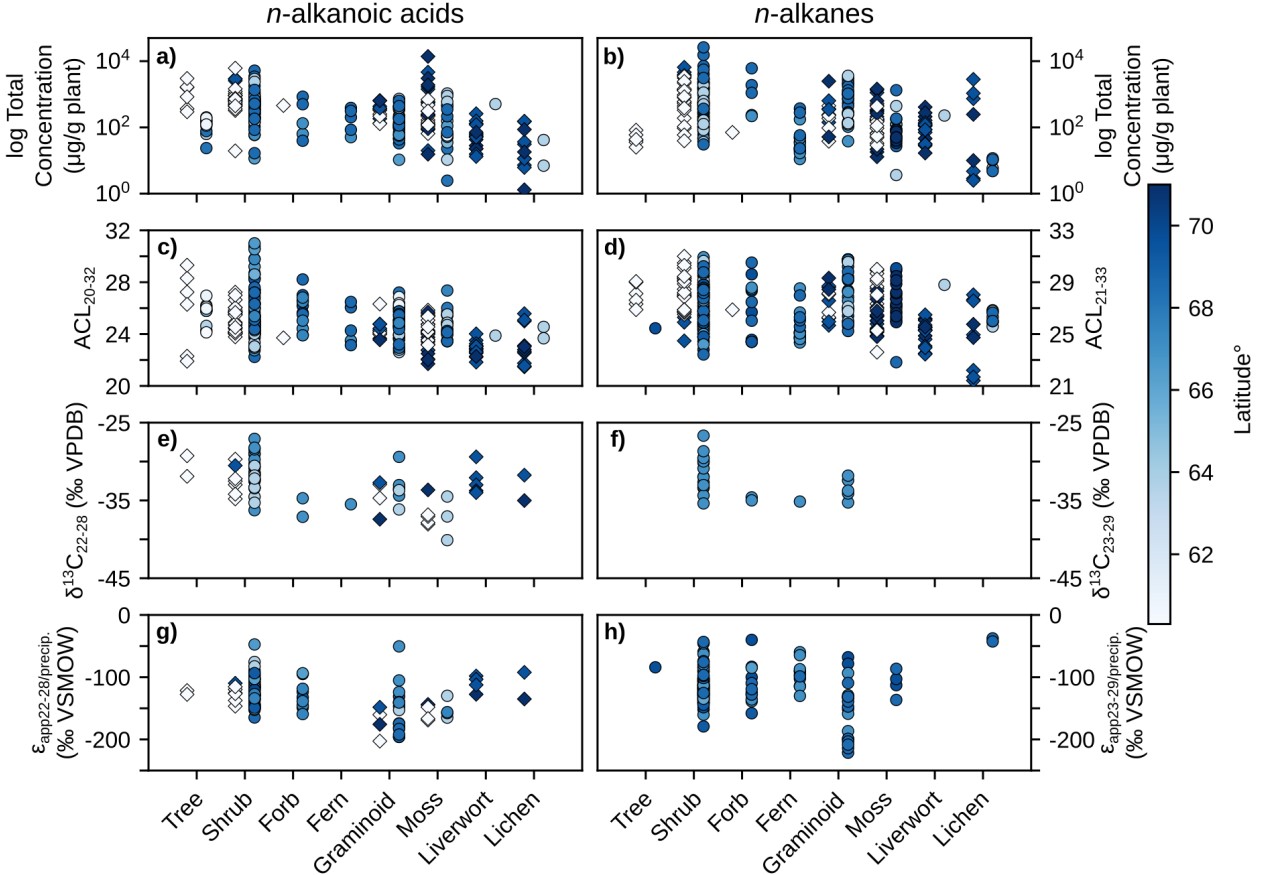

**Figure 3.** Scatterplots of plant wax results from this study (diamonds) and the pan-Arctic data compilation (circles) grouped by plant growth form. Left column panels (a, c, e, g) contain *n*-alkanoic acid results, right column panels (b, d, f, h) contain *n*-alkane results. (a-b) Log total plant wax concentration, (c-d) Plant wax ACL, (e-f) Plant wax $\delta^{13}$C, (g-h) Plant wax $\epsilon_{app}$ relative to MAF precipitation amount-weighted $\delta^2$H. Data points are shaded by sampling site latitude.

S1a) and from 0.8 to 44.8 with an average of $5.0 \pm 6.6$ for *n*-alkanes (Fig. S1b). Trees and lichens had both the highest mean CPI values and variability in both the new and compiled data, while liverworts had the lowest CPI values overall. Within the pan-Arctic data compilation, all *n*-alkanoic acid samples had CPI values $\geq 1$ and 98% were $\geq 2$, while 93% of *n*-alkane CPI values were $\geq 1$ and 74% were $\geq 2$.

### 3.2 Plant Wax $\delta^{13}$C

We measured even-chain *n*-alkanoic acid (C$_{22}$ through C$_{28}$) $\delta^{13}$C in 25 ECA plants. Here, we refer to $\delta^{13}$C results as the concentration-weighted average of those chain-lengths (C$_{23}$ through C$_{29}$ for *n*-alkanes in other datasets). ECA *n*-alkanoic acid $\delta^{13}$C ranged from -38.1 to -29.3‰ with an average of $-33.5 \pm 2.5‰$ (Fig. 3e). Pan-Arctic *n*-alkanoic acid $\delta^{13}$C ranged from



**Figure 4.** Similarities of plant wax data between plant growth forms. Matrices of Mann-Whitney U test p-values between pan-Arctic plant wax indices of major plant growth forms (Tre = trees, Shr = shrubs, For = forbs, Fer = ferns, Gra = graminoids, Mos = mosses, Liv = liverworts, Lic = lichens). Left column panels (a, c, e, g) use plant wax $n$-alkanoic acid data and right column panels (b, d, f, h) use plant wax $n$-alkane data. (a-b) Total plant wax concentration ($\mu$g/g plant). (c-d) Plant wax ACL. (e-f) Plant wax $\delta^{13}$C. (g-h) Plant wax $\epsilon_{app}$ relative to MAF precipitation amount-weighted $\delta^2$H. Green cells represent $p$-values < 0.05, indicating the two datasets are significantly different from each other, grey cells represent $p$-values $\geq$ 0.05, and white cells indicate that there was an insufficient number of observations to perform the test.

-40.1 to -27.1‰ with an average of -33.0 ± 2.6‰ and $n$-alkane $\delta^{13}$C ranged from -35.4 to -26.7‰ with an average of -32.2 ± 2.3‰. The range in ECA $\delta^{13}$C between chain-lengths within an individual sample was an average 3.1 ± 2.3‰. In both the



ECA and pan-Arctic *n*-alkanoic acid datasets, trees were the most $^{13}$C-enriched and mosses were the most $^{13}$C-depleted. The pan-Arctic data compilation included *n*-alkane $\delta^{13}$C only for shrubs, forbs, ferns, and graminoids. Among these four growth forms, shrubs were the most $^{13}$C-enriched and ferns were the most $^{13}$C-depleted (Fig. 3f). Shapiro-Wilk tests were limited by

many growth forms not having enough measurements ($n < 3$) in each compound class. All growth forms that did have enough measurements were normally distributed (Table S1). Shrub and moss *n*-alkanoic acid $\delta^{13}$C were the most significantly different from other plant growth forms (Fig. 4e). No growth forms had significantly different distributions of *n*-alkane $\delta^{13}$C (Fig. 4f).

### 3.3 Plant Wax $\delta^2$H and $\epsilon_{app}$

We measured even-chain *n*-alkanoic acid $\delta^2$H, across the same chain-length range as $\delta^{13}$C, in 24 ECA plants. One sample that
was first analyzed for $\delta^{13}$C did not have sufficient plant wax material to also measure $\delta^2$H. Again, we refer to the $\delta^2$H and $\epsilon_{app}$ results as the chain-length concentration-weighted average value in each plant sample. The individual sample range in ECA *n*-alkanoic acid $\delta^2$H between chain-lengths was an average of 30.5 ± 18.6‰. In the pan-Arctic dataset, we found that *n*-alkanoic acid $\delta^2$H had weak, positive correlation ($r = 0.25$; $p = 0.02$) with MAF precipitation $\delta^2$H (Fig. S2a), while n-alkane $\delta^2$H had a moderate, positive correlation ($r = 0.64$; $p < 0.01$) with MAF precipitation $\delta^2$H (Fig. S2b). When removing the
data from Hollabåttjønnen Bog in northern Norway (Balascio et al., 2018), however, the correlation between *n*-alkane $\delta^2$H and precipitation $\delta^2$H became weakly negative and not statistically significant ($r = -0.18$; $p = 0.13$). Additionally, the two plant wax compound-specific $\delta^2$H datasets do not represent the same samples/sampling locations, with the lack of *n*-alkane $\delta^2$H from ECA plants as an example.

Since $\delta^2$H values of individual plants are partially determined by the $\delta^2$H of precipitation at each site, we normalize that
large amount of variability in our datasets by comparing the $\epsilon_{app}$ values between plant growth forms, instead. ECA *n*-alkanoic acid $\epsilon_{app}$ ranged from -202.7 to -92.2‰ with an average of -135.7 ± 25.9‰ (Fig. 3g). Pan-Arctic *n*-alkanoic acid $\epsilon_{app}$ ranged from -202.7 to -47.4‰ with an average of -130.2 ± 27.2‰, and *n*-alkane $\epsilon_{app}$ ranged from -221.4 to -37.6‰ with an average of -117.0 ± 31.1‰. Graminoids had the most negative *n*-alkanoic acid $\epsilon_{app}$ values in both the ECA and pan-Arctic datasets. Liverworts and lichens had the least negative *n*-alkanoic acid $\epsilon_{app}$ values in the ECA dataset, which were also the only such
samples of those growth forms in the pan-Arctic. This pattern of graminoids and lichens (no liverwort data available) having the most and least negative $\epsilon_{app}$ values, respectively, was also true for pan-Arctic *n*-alkanes (Fig. 3h). Similar to $\delta^{13}$C, we were unable to perform Shapiro-Wilk and Mann-Whitney U tests on the $\delta^2$H of some growth forms due an insufficient number of measurements (Table S1, Figure 4). Most growth forms with enough $\epsilon_{app}$ measurements were found to be normally distributed except for shrubs, despite having the most measurements available among all growth forms (Fig. 3g, h). Moss and lichen
$\epsilon_{app}$ distributions were the most significantly different from other growth forms in the *n*-alkanoic acid (Fig. 4g) and *n*-alkane (Fig. 4h) datasets, respectively. However, Mann-Whitney U tests with stable isotope data more often showed that measurement distributions within individual plant growth forms were not significantly different compared to similar tests with total plant wax concentration and ACL data.





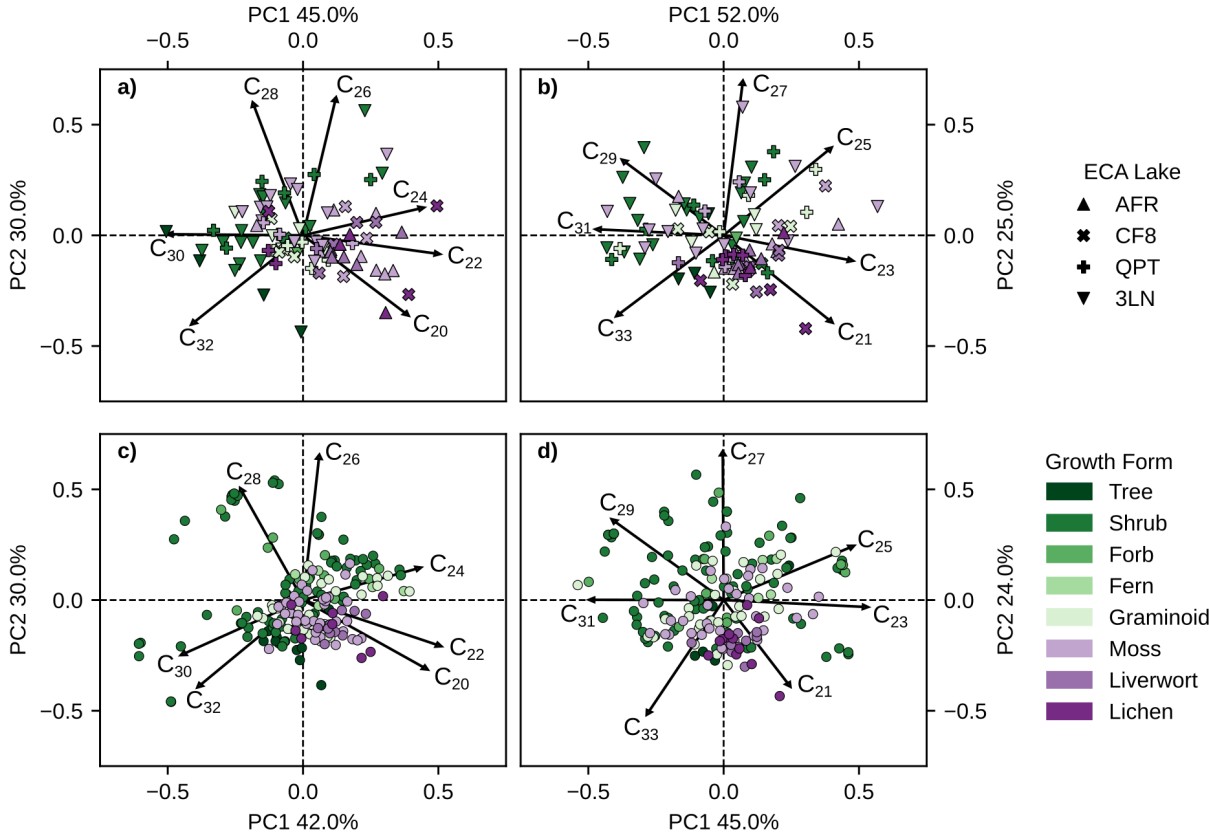

**Figure 5.** PCA biplots (principal components 1 and 2) generated from clr-transformed relative abundances of different chain-lengths of (a) ECA *n*-alkanoic acids, (b) ECA *n*-alkanes, (c) pan-Arctic *n*-alkanoic acids, (d) pan-Arctic *n*-alkanes. Color legend represents major plant growth forms, vascular plants are green shades, non-vascular plants are purple shades. Plot shapes in panels a and b denote each ECA lake. Multiple samples of the same species from the same site are averaged.

## 3.4 Principal Component Analysis

We performed PCA on plant wax *n*-alkanoic acid and *n*-alkane chain-length abundances using both the ECA lakes dataset and pan-Arctic data (Fig. 5). The first two principal components (PCs) in all analyses explained the majority of overall variance, ranging from 69 to 77% (Fig. S3a-d). The explained variance successively decreased substantially for the remaining PCs, 3-7, with each explaining no more than 14% of the total variance. Eigenvectors for the four shortest chain-length plant waxes ($C_{20}$ to $C_{26}$/$C_{21}$ to $C_{27}$; *n*-alkanoic acids/*n*-alkanes) always had positive loadings on PC1 (x-axis), and eigenvectors for the

three longest chain-length plant waxes ($C_{28}$-$C_{32}$/$C_{29}$-$C_{33}$) had negative loadings. Generally, eigenvectors for the one to two shortest and longest chain-length had negative loadings on PC2 (y-axis) and eigenvectors for the remaining chain-length plant waxes had positive loadings. For interpreting the PCA scores of individual samples, PC1 scores reflect a greater abundance





of mid-chain (positive) vs. long-chain (negative) waxes, and negative PC2 scores represent a greater abundance of the longest ($C_{32}/C_{33}$) and/or shortest ($C_{20}/C_{21}$) chain-lengths.

ECA vascular plants tended to produce more longer-chain *n*-alkanoic acids than non-vascular plants, resulting in them plotting more negatively on PC1 (Fig. 5a). This relationship partly covaries with sampling location, since non-vascular plants are more prevalent in the northern ECA sites (AFR, CF8) while vascular plants dominate southern ECA sites (QPT, 3LN). Pan-Arctic vascular plant *n*-alkanoic acid distributions were more variable, but were mostly absent from Quadrant IV containing the $C_{20}$ and $C_{22}$ loading factors (Fig. 5c). The majority of non-vascular pan-Arctic *n*-alkanoic acids had positive PC1 scores

associated with shorter-chain length production and negative PC2 scores. ECA vascular plant *n*-alkanes also tended to have negative PC1 scores, while moss PC scores were highly variable (Fig. 5b). ECA liverwort and lichen *n*-alkane distributions were more tightly clustered than *n*-alkanoic acids, mostly plotting along the $C_{21}$ loading factor. This pattern of high PC score variability among *n*-alkane distributions for all plant growth forms, except for liverworts and lichens, was also true for pan-Arctic *n*-alkane distributions (Fig. 5d).

### 290   3.5   Plant Waxes and Environmental Parameters

The sampling locations in the new ECA dataset and pan-Arctic data compilation spanned a substantial range in MAF (growing season) environmental parameters across 14° latitude. Temperatures ranged from 3.6 to 12.4 °C (Fig. 2a), total precipitation ranged from 130.3 to 803.5 mm (Fig. 2b), relative humidity ranged from 56.3 to 80.1% (Fig. 2c), elevation ranged from 0 to 950 masl (Fig. 2d), and precipitation $\delta^2$H ranged from -145.5 to -86.6‰ (Fig. 2e). The transect of ECA sites (AFR, CF8, QPT, 3LN)

also contained a wide range of precipitation $\delta^2$H values, but are generally characterized as being colder and drier compared to other sampling sites in the pan-Arctic dataset. We used site temperature, precipitation amount, relative humidity, and elevation in Pearson correlation tests with plant wax data. Correlations between individual environmental parameters (i.e. temperature vs. precipitation) were mostly weak ($|r| \leq 0.4$), with the only correlation between temperature and relative humidity ($r$ = -0.50) being of moderate ($0.4 < |r| \leq 0.8$) strength (Fig. 6a, b; outside of the black boxes).

Pearson correlations between plant wax *n*-alkanoic acid/*n*-alkane data from all plant growth forms and environmental parameters yielded weak to moderate positive and negative linear relationships (Fig. 6a, b; black boxes), but with many of them not being statistically significant ($p > 0.05$). The only significant relationships with *n*-alkanoic acid data were weak positive ($r$ = 0.16, 0.19) correlations between total (log) concentration and precipitation amount and relative humidity and weak to moderate positive ($r$ = 0.36, 0.42) correlations between ACL and temperature and precipitation amount (Fig. 6a). There were

no significant correlations between *n*-alkanoic stable isotope data and environmental parameters. Terrestrial plant *n*-alkane had a weak positive ($r$ = 0.17) and weak negative ($r$ = -0.18) correlations between ACL and precipitation amount and elevation, respectively (Fig. 6b). *n*-alkane $\epsilon_{app}$ also had a weak positive ($r$ = 0.37) correlation with precipitation amount. The $\delta^{13}$C values of *n*-alkanes were only reported in plants from a single study in west Greenland sampled during a single year (2015) (Dion-Kirschner et al., 2020), which prevented comparisons with environmental parameters.

We repeated these Pearson correlations with the plant wax data of a single growth form removed each time (e.g., assessed correlations with all trees removed, then all shrubs removed, etc). This allowed us to assess the influence of different growth





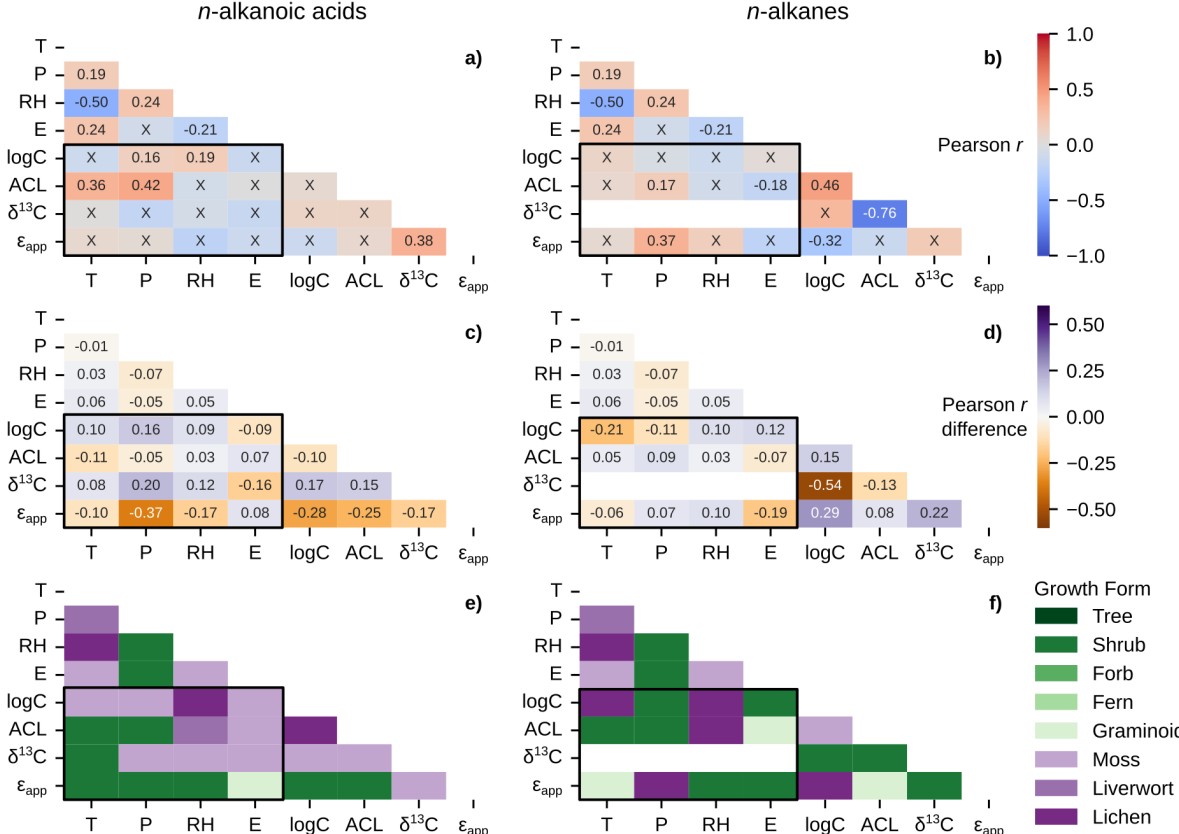

**Figure 6.** Matrices of Pearson correlation results using environmental parameters (T = MAF temperature; P = MAF precipitation amount; RH = MAF relative humidity; E = elevation) and chain-length amount-weighted plant wax indices, with black boxes highlighting the correlation tests between the two. Left column panels (a, c, e) use plant wax *n*-alkanoic acid data and right column panels (b, d, f) use plant wax *n*-alkane data. (a-b) Pearson correlation *r* values with all plant growth forms included. "X" annotations indicate the correlation is not significant ($p > 0.05$). (c-d) Maximum, positive or negative, Pearson *r* value difference between the full dataset and tests removing one plant growth form at a time. (e-f) Plant growth form removed that is responsible for the values in panels c and d.

forms on each correlation via the change in magnitude and direction of the *r*-value each time (Fig. 6c, d) while tagging the growth form that caused the greatest change when removed (Fig. 6e, f). Pearson *r*-value changes ranged from -0.37 to 0.20 for the *n*-alkanoic acid dataset (Fig. 6c), with the removal of mosses and shrubs being responsible for each extreme, respectively (Fig. 6e). These two growth forms were most often responsible for the greatest change in *r*-value for plant wax vs. environmental correlations. Changes in *n*-alkane correlation *r*-values ranged from -0.21 to 0.12 (Fig. 6d) with lichens and shrubs being responsible for the extreme values and the most frequent changes in *r*-value (Fig. 6f). Liverwort and lichen removal also caused some of the largest *r*-value changes in the *n*-alkanoic acid dataset, while graminoid removal did the same for the *n*-alkane dataset.



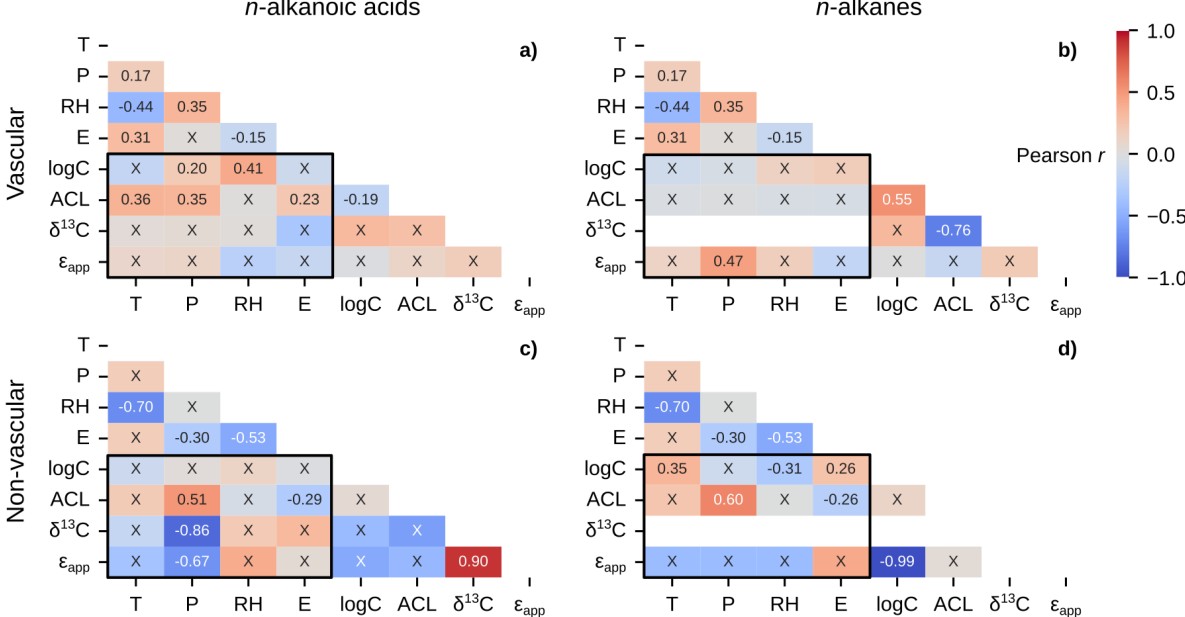

**Figure 7.** Matrices of Pearson correlation results using environmental parameters (T = MAF temperature; P = MAF precipitation amount; RH = MAF relative humidity; E = elevation) and chain-length amount-weighted plant wax indices, with black boxes highlighting the correlation tests between the two. Left column panels (a, c) use plant wax $n$-alkanoic acid data and right column panels (b, d) use plant wax $n$-alkane data. (a-b) Vascular plants (trees, shrubs, forbs, ferns, graminoids). (c-d) Non-vascular plants (mosses, liverworts, lichens). "X" annotations indicate the correlation is not significant ($p > 0.05$).

To further dissect potential empirical relationship between plant wax data and environmental parameters, we investigate linear relationships between several subsets of the pan-Arctic data, including vascular and non-vascular plants (Fig. 7), individual plant growth forms (Fig. 8), and individual plant genera/species (Fig. 9). For all data subsets listed, the majority of correlations were not statistically significant ($p > 0.05$), due to either high variability (very low $r$-value) in the plant wax data, low sample sizes, or a combination of the two.

Vascular plants reflected a similar pattern in correlations to the full dataset: Weak to moderate positive correlations with $n$-alkanoic acid concentration and ACL (Fig. 7a), and a moderate correlation between $n$-alkane $\epsilon_{app}$ and precipitation amount (Fig. 7b). Non-vascular plant waxes revealed different relationships with environmental parameters. Both $n$-alkanoic acid and $n$-alkane ACL was moderately positively ($r = 0.51, 0.60$) correlated with precipitation amount, whereas $n$-alkanoic acid $\delta^{13}$C and $\epsilon_{app}$ were both moderately to strongly negatively ($r = -0.67, -0.86$) correlated with precipitation amount (Fig. 7c, d).

Shrubs, graminoids, and mosses contained the most plant wax measurements among plant growth forms, and spanned the greatest number of sampling locations/unique environmental data points. The shrub-only data subset, containing the most measurements among growth forms, maintained the correlation patterns present in the full pan-Arctic and vascular plant groupings but with a now-weak positive correlation between $n$-alkane $\epsilon_{app}$ and precipitation amount (Fig. 8a, b). Graminoids displayed a





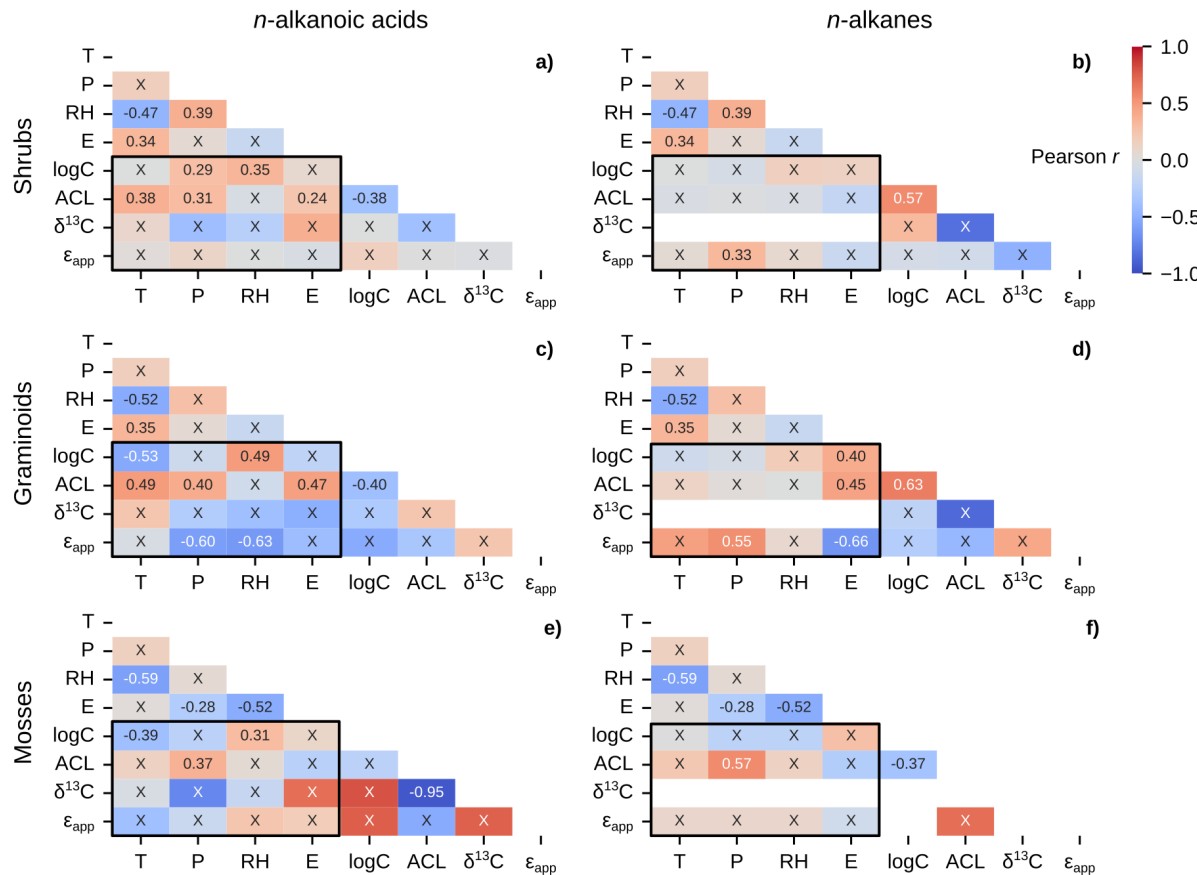

**Figure 8.** Matrices of Pearson correlation results using environmental parameters (T = MAF temperature; P = MAF precipitation amount; RH = MAF relative humidity; E = elevation) and chain-length amount-weighted plant wax indices of individual plant growth forms, with black boxes highlighting the correlation tests between the two. Left column panels (a, c, e) use plant wax $n$-alkanoic acid data and right column panels (b, d, f) use plant wax $n$-alkane data. (a-b) Shrubs. (c-d) Graminoids. (e-f) Mosses. "X" annotations indicate the correlation is not significant ($p > 0.05$).

number of moderate correlations not seen in other data subsets, including relationships between $\epsilon_{app}$ with precipitation amount

($r$ = -0.60, -0.55; $n$-alkanoic acids, $n$-alkanes), relative humidity ($r$ = 0.63; $n$-alkanoic acids), and elevation ($r$ = -0.66; $n$-alkanes), along with a negative ($r$ = -0.53) relationship between total $n$-alkanoic acid total concentration and temperature (Fig. 8c, d). Mosses also had a weakly negative ($r$ = -0.39) correlation between $n$-alkanoic acid total concentration and temperature, along with weakly positive correlations between total concentration and relative humidity ($r$ = 0.31) and ACL and precipitation amount ($r$ = 0.37) (Fig. 8e). The only significant correlation present in moss $n$-alkanes was a moderately positive ($r$ = 0.57)

correlation between ACL and precipitation amount (Fig. 8f).





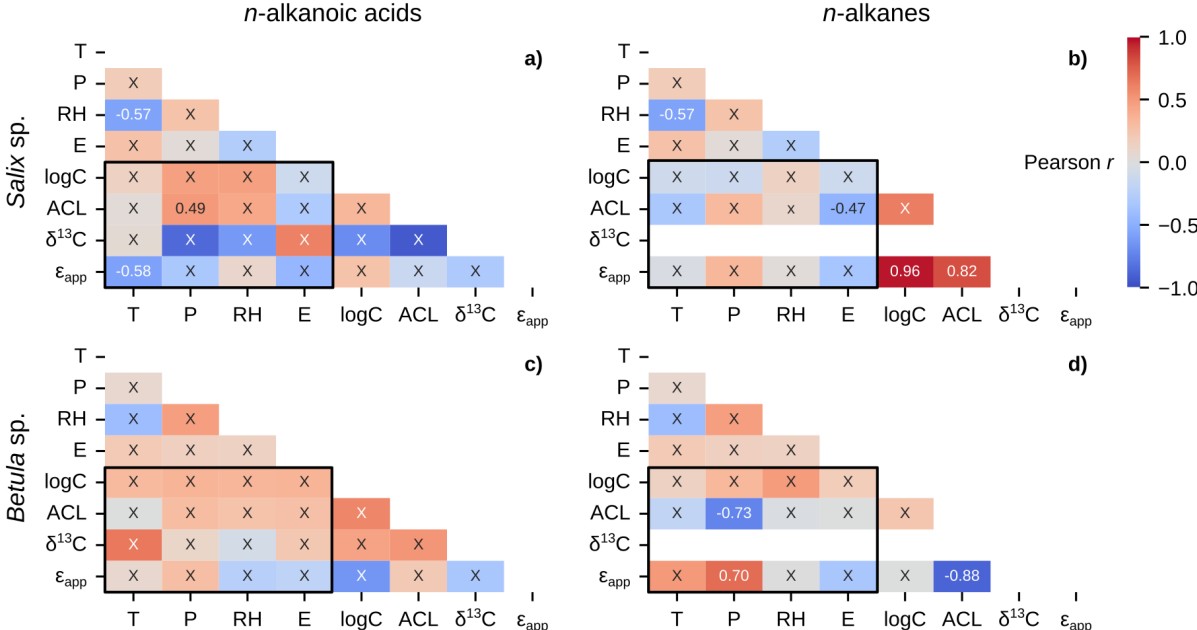

**Figure 9.** Matrices of Pearson correlation results using environmental parameters (T = MAF temperature; P = MAF precipitation amount; RH = MAF relative humidity; E = elevation) and chain-length amount-weighted plant wax indices of individual plant genera/species, with black boxes highlighting the correlation tests between the two. Left column panels (a, c) use plant wax $n$-alkanoic acid data and right column panels (b, d) use plant wax $n$-alkane data. (a-b) All plants within the *Salix* genus only. (c-d) *Betula* genus only. "X" annotations indicate the correlation is not significant ($p > 0.05$).

*Betula* sp. contained the greatest number of measurements and unique sampling locations among all Arctic plant genera, and all but one (56 of 57) samples were from the *Betula nana/glandulosa* complex. We excluded the *Betula pubescens* sample from Hollabåttjønnen Bog (Balascio et al., 2018) in the correlations due to it being classified as a tree, not a shrub like *Betula nana* and *Betula glandulosa*. *Salix* sp. was the second most sampled genus, but contained a more diverse set of species (e.g., *Salix*

*glauca*, *Salix arctica*, *Salix pulchra*, etc.). The only significant relationships between *Salix* sp. and environmental parameters were a moderate positive ($r = 0.49$) correlation between $n$-alkanoic acid ACL and precipitation amount, a moderately negative ($r = -0.58$) correlation between $n$-alkanoic acid and $\epsilon_{app}$ temperature, and a moderately negative ($r = -0.47$) correlation between $n$-alkane ACL and elevation (Fig. 9a, b). There were no significant correlations between *Betula* sp. $n$-alkanoic acids and any environmental parameters (Fig. 9c). *Betula* sp. $n$-alkane ACL and $\epsilon_{app}$ were moderately negatively ($r = -0.73$) and moderately

positively ($r = 0.70$) correlated with precipitation amount (Fig. 9d).





## 4   Discussion

In this section, we evaluate three facets of the results from each plant wax data type (total concentration, chain-length distribution, $\delta^{13}$C, $\delta^2$H/$\epsilon_{app}$): 1) environmental correlations and their relationship to previously hypothesized mechanisms influencing plant wax production and stable isotope fractionation, 2) effects of plant growth forms, and 3) implications for using this proxy
for Arctic paleoclimate reconstructions.

Weak correlations, overall, between environmental parameters across pan-Arctic sampling locations indicate that individual environmental controls on plant wax indices can be evaluated without much covarying influence of others (i.e. the effects of precipitation amount vs. temperature on ACL values). However, correlation tests with the removal of a single plant growth form reveal that some correlation strengths may be largely driven by the sample size and location of certain growth forms
(Fig. 6e, f). Shrubs contain the greatest number of measurements and span the most unique sampling locations, so it is not surprising that their presence/absence has a greater effect on larger data subsets (vascular, all data). Mosses, liverworts, and lichens are more abundant in bioclimates A-C, which skews their sampling towards colder, drier sampling locations. Therefore, their presence/absence may have an "anchoring effect" on the cold/dry end of these environmental gradients.

Performing large amounts of independent significance tests, 28 in each half-matrix for a given data subset for a single plant
wax compound class, also has an inherent potential to yield some false positives. By using a 95% confidence interval ($p$ = 0.05), we expect that each set of tests contains one or two false positives, or about one in the subset of 16 tests between plant wax indices and environmental parameters. It is possible, therefore, that some comparisons that only produced one significant correlation between plant wax indices and environmental parameters, including vascular plant (Fig. 7b), shrub (Fig. 8b), moss (Fig. 8f), and *Salix* sp. (Fig. 9b) *n*-alkanes actually did not yield any significant correlations when accounting for the likelihood
of false positives. In contrast, comparisons with two or more significant correlations are highly likely (>95% confidence) to reflect real patterns. In the following sections, therefore, we interpret significant correlations only where there are previously documented physiological and/or environmental mechanisms driving plant wax production.

### 4.1   Total Plant Wax Concentration

Plant wax production is often a response mechanism by individual plants to prevent water loss under stress (Burlett et al.,
2025; Lewandowska et al., 2020): more water stress causes a higher total concentration of plant waxes. Hoffmann et al. (2013) demonstrated that total *n*-alkane concentrations in *Eucalyptus* sp. were negatively correlated with annual precipitation amount and relative humidity. However, we generally do not observe this relationship between terrestrial Arctic plants and MAF precipitation amount or relative humidity. Cold summer temperatures and restricted soil drainage due to bedrock and permafrost minimize water limitations in the Arctic (Gold and Bliss, 1995). The only exception was non-vascular plant *n*-
alkanes, which had a weak negative correlation with relative humidity (Fig. 7d). In fact, the environmental correlations of moderate strength with total concentration were actually positive, as seen in vascular plant ($r$ = 0.41) and graminoid ($r$ = 0.49) *n*-alkanoic acid data subsets with relative humidity (Fig. 7a; Fig. 8c). It is possible that any trends in total plant wax production





are masked by the high measurement variability present in all plant growth forms: the average standard deviation for growth form-specific *n*-alkanoic acid concentrations was 572 $\mu$g/g and 891 $\mu$g/g for *n*-alkanes.

Shapiro-Wilk tests showing that total plant wax concentrations within individual plant growth forms are often log-normally distributed (Table S1) agrees with other large compilations of plant wax data from western Africa and the Tibetan Plateau (Garcin et al., 2014; Yang and Bowen, 2022). Some studies have demonstrated significant differences in total plant wax concentrations between plant types, notably that angiosperms produce more *n*-alkanes than gymnosperms (Bush and McInerney, 2013; Diefendorf et al., 2011). However, fewer such comparisons have been performed between the groupings of plant growth

forms analyzed in this study. Shrubs have distributions of total *n*-alkanoic acid and *n*-alkane concentrations that are significantly different (Fig. 4a, b) and greater (Fig. 3a, b) than most other plant growth forms. This supports the hypothesis that the waxes in these plants have greater potential to be strongly represented in Arctic sedimentary records (Dion-Kirschner et al., 2020; Hollister et al., 2022). The opposite may be the case for growth forms that produce relatively low plant wax concentration, including liverworts and lichens.

## 395   **4.2   Chain-Length Distributions**

The mechanisms behind a plant's production of different plant wax carbon chain-lengths in response to environmental conditions is much less understood compared to the total production of these compounds. It has been suggested that plants produce longer chain-lengths, resulting in a greater ACL value, to combat water stress in warmer and/or drier growing conditions (Shepherd and Wynne Griffiths, 2006). These trends have been observed in plant *n*-alkane ACL across substantial temperature and

aridity gradients in the eastern U.S. (Tipple and Pagani, 2013), central U.S. (Bush and McInerney, 2015), and Europe (Sachse et al., 2006). Therefore, we would expect terrestrial Arctic plant wax ACL to be positively correlated with temperature and negatively correlated with precipitation amount and relative humidity.

    We do observe weak to moderate positive correlations between ACL and temperature in all plant *n*-alkanoic acids ($r = 0.36$; Fig. 6a), vascular plants ($r = 0.36$; Fig. 7a), shrubs ($r = 0.38$; Fig. 8a), and graminoids ($r = 0.49$; Fig. 8c), but not in any

*n*-alkane data subsets. Nearly all significant correlations between ACL and precipitation amount and relative humidity were weakly to moderately positive, with the exception of *Betula* sp. *n*-alkane ACL ($r = -0.73$; Fig. 9d). Apart from that moderate correlation in *Betula* sp., the lack of significant correlations, otherwise, supports previous findings on *Betula* sp. that their plant wax distributions are highly variable but not governed by environmental variables (Weber and Schwark, 2020). While these results disagree with the assumed mechanisms behind plant wax chain-length production, Tipple and Pagani (2013) and Bush

and McInerney (2015) also do not find ACL to have a significant relationship with water availability. Hoffmann et al. (2013) also showed that the ACL for different genera can have opposing positive and negative correlations to precipitation and relative humidity within the same environmental gradient. Such an effect may obfuscate plant wax relationships in individual taxa to environmental data when data are grouped at the growth form level and above.

    Mann-Whitney U tests revealed that ACL values did not differ between vascular plant growth forms, while non-vascular

plants, particularly liverworts and lichens, had unique ACL distributions relative to all other growth forms (Fig. 4c, d). These patterns were also seen in the PCA biplot space where non-vascular plants tended to cluster in Quadrant IV with the two





shortest *n*-alkanoic acid ($C_{20}$, $C_{22}$) and *n*-alkane ($C_{21}$, $C_{23}$) chain-lengths (Fig. 5c, d). Trees, which had some of the highest average ACL values, mostly plotted in Quadrant III of the PCA biplots containing the two longest chain-lengths. This is in contrast to other sampled temperate and subtropical/arid biomes where tree ACL was shown to be lower than many shrubs and graminoids (Freimuth et al., 2019; Howard et al., 2018). Bush and McInerney (2013) found similar ACL patterns in *n*-alkanes from temperate plant species where vascular plant species had similar, highly variable ACL distributions while mosses had distinctly lower values. Interestingly, pan-Arctic moss *n*-alkane ACL values were not significantly different from vascular plants, whereas the Bush and McInerney (2013) trend held true for pan-Arctic *n*-alkanoic acids. These patterns in Arctic vegetation echo warnings from previous compilations of plant ACL values that chain-length distributions are not sufficient to fingerprint different types of vascular plants (Bush and McInerney, 2013; Liu et al., 2022), but discerning changes between vascular and non-vascular plant wax sources may still be possible.

### 4.3  $\delta^{13}$C

Fractionation of stable carbon isotopes between the atmosphere and $C_3$ plant waxes is strongly influenced by a plant's stomatal conductance in its leaves responding to water availability (Farquhar et al., 1982). In global compilations of plant samples, this mechanism is illustrated as negative correlations between plant wax *n*-alkane $\delta^{13}$C and temperature and precipitation amount (Diefendorf et al., 2010; Liu et al., 2022; Wang et al., 2025). Arctic plant wax *n*-alkanoic acid and *n*-alkane $\delta^{13}$C values all fell within the global range of variability ($\sim$-45 to -20‰) for $C_3$ photosynthesizers (Diefendorf and Freimuth, 2017; Liu et al., 2022). However, the only significant correlation between $\delta^{13}$C and any environmental variable was a strongly negative ($r =$ -0.86) relationship between non-vascular *n*-alkanoic acids and precipitation amount (Fig. 7c). Interestingly, mosses are the only growth form of the non-vascular plants that can have stomata within their structure, while liverworts and lichens do not (Renzaglia et al., 2020). Therefore, non-vascular plant $^{13}$C fractionation sensitivity to water availability in these growth forms is likely driven by other physiological factors.

In this analysis, the Arctic plant wax $\delta^{13}$C were limited to this study, Lake QPT in the ECA (Hollister et al., 2022), and west Greenland (Dion-Kirschner et al., 2020), so it is possible that significant correlations in vascular plants were not resolved due to a low measurement count across a smaller gradient of temperature and precipitation amount. This also applies to investigating differences in $\delta^{13}$C between plant growth forms. Relatively $^{13}$C-enriched shrubs and $^{13}$C-depleted mosses were the most significantly different groups, but many growth forms have three or fewer site and species-averaged measurements (Fig. 3e, f). The high degree of overlap in $\delta^{13}$C values between plant growth forms suggests that caution should be used when interpreting past terrestrial vegetation changes using sedimentary plant wax $\delta^{13}$C in the Arctic. This large overlap impacts the ability to construct unique $\delta^{13}$C-based vegetation endmembers for numerical modeling of plant wax sources (Yang and Bowen, 2022; Yu et al., 2024).

### 4.4  $\delta^2$H and $\epsilon_{app}$

Plant wax $\delta^2$H values and their associated $\epsilon_{app}$ relative to their source water $\delta^2$H are controlled by multiple factors, including evaporative $^2$H-enrichment that occurs in the soil and plant's xylem prior to plant wax synthesis, as well as the biosynthetic




fractionation factor ($\epsilon_{bio}$) which heavily discriminates against the heavier $^2$H isotope (Sachse et al., 2012). O'Connor et al. (2020) found moderate positive correlations between $\epsilon_{bio}$ and temperature and precipitation amount along a north-south transect of Alaskan shrubs and forbs. However, disentangling the environmental controls on $^2$H fractionation pre- (evaporation) and post- ($\epsilon_{bio}$) plant wax synthesis would require more measurements of plant leaf water, which are currently limited to only two other Arctic datasets/sampling locations (Berke et al., 2019; Daniels et al., 2017) in addition to the data in O'Connor et al. (2020).

If we assume $\epsilon_{bio}$ is constant for individual plants, increased evaporation results in a smaller absolute (less negative) $\epsilon_{app}$ value (Sachse et al., 2006). In this framework, environmental correlations with $\epsilon_{app}$ values are expected to relate to evaporation potential: positive correlations with increasing temperature and negative correlations with increasing precipitation and relative humidity. However, the only cases in which these expected relationships were present in the pan-Arctic dataset were a moderately negative correlation between non-vascular $n$-alkanoic acid $\epsilon_{app}$ and precipitation amount ($r$ = -0.67; Fig. 7c) as well as moderately negative correlations between graminoid $n$-alkanoic acid $\epsilon_{app}$ and precipitation amount ($r$ = -0.60) and relative humidity ($r$ = -0.63; Fig. 8c). A number of data subsets expressed opposite relationships than expected, including moderately positive correlations between $n$-alkane $\epsilon_{app}$ in vascular plants ($n$ = 0.47; Fig. 7c), graminoids ($r$ = 0.55; Fig. 8d), and *Betula* sp. ($r$ = 0.70; Fig. 9d). Liu et al. (2023) found weakly negative correlations between mean annual precipitation amount and terrestrial monocot ($r$ = -0.36) and dicot ($r$ = -0.38) $n$-alkane $\epsilon_{app}$ across China. However, mean annual precipitation amount in their sampling locations ranged from 30 to 1720 mm, with much more scatter in $\epsilon_{app}$ values up to ~600 mm. It is possible that the ranges of MAF precipitation amount, temperature, and relative humidity in the pan-Arctic dataset are not enough to drive consistent, significant changes in $\epsilon_{app}$ (Feakins and Sessions, 2010).

Weak correlations between plant wax and precipitation $\delta^2$H illustrates the high variability in $\epsilon_{app}$ at each sampling location (Fig. S2). The anomalously $^2$H-enriched values from Hollabåttjønnen Bog in northern Norway (Balascio et al., 2018) were likely caused by those plants using bog water (-60‰) that was much more $^2$H-enriched than our calculated amount-weighted MAF precipitation $\delta^2$H (-87‰) based on ERA5 reanalysis and the Online Isotopes in Precipitation calculator (Bowen and Revenaugh, 2003; Bowen et al., 2005; Hersbach et al., 2020). While such issues can occur when using this approach, it serves as a consistent data source and methodology for datasets where site-specific meteorological and precipitation isotope data were not collected nor available elsewhere. For example, this information was not collected at each ECA lake and there are large discrepancies in distance to the nearest weather and precipitation isotope monitoring station (Gorbey et al., 2022). Additionally, Saishree et al. (2023) found high $\epsilon_{app}$ variance (standard deviation > 20‰) in deciduous and evergreen trees growing in the same climate and irrigated with $\delta^2$H-controlled water. Therefore, high $\epsilon_{app}$ variability within individual growth forms may also be a function of plant physiology, not just from discrepancies between measured and inferred plant source water.

The lack of distinct distributions of $\epsilon_{app}$ values between plant growth forms may be beneficial for reconstructions of past Arctic hydrology. The only pattern consistent with other, globally-distributed plant wax $\epsilon_{app}$ compilations was Arctic graminoids have the most negative values, on average, among growth forms (Gao et al., 2014; Sachse et al., 2012). Some studies report vegetation-corrected source water $\delta^2$H records based on $\epsilon_{app}$ values of different plant types and their estimated contributions to the sediment (Feakins, 2013; Holtzman et al., 2025). Based on our results, calculating MAF precipitation $\delta^2$H using pan-Arctic




$\epsilon_{app}$ averages of -130.2 ± 27.2‰ for *n*-alkanoic acids and -117.0 ± 31.1‰ for *n*-alkanes may be sufficient in the absence of a paired vegetation record from the same site.

## 5   Conclusions

The goal of this study was to evaluate whether variability in terrestrial Arctic plant waxes is driven by environmental factors or physiological differences between plant growth forms. Overall, terrestrial Arctic plant waxes do not exhibit substantial
empirical relationships to any of the environmental parameters tested in this study. The majority of Pearson correlations were not statistically significant. Those that were significant were mostly classified as weak ($|r| \leq 0.4$) and often contradicted previously proposed environmental drivers on plant wax data. A fundamental assumption of plant wax-based paleoclimate reconstructions is that each proxy is primarily recording one type of environmental or ecological change: e.g., changes in ACL over time represent changes in local vegetation composition, changes in $\delta^2$H represent changes in precipitation $\delta^2$H.
Weak correlations suggest that the environmental ranges captured across the modern Arctic do not exert a strong influence on how plants synthesize different plant wax chain-lengths or fractionate stable carbon and hydrogen isotopes. Differences in plant growth forms appeared to affect the total concentration and chain-length distributions of plant waxes, while stable isotope $\delta^{13}$C and $\epsilon_{app}$ variability were much more consistent between plant growth forms. Therefore, changes in plant wax distributions in paleoclimate records do likely reflect changes in terrestrial plant taxa present over time, and past changes in
plant wax $\delta^2$H likely reflect changes in source water (precipitation) $\delta^2$H, both without a strong, confounding influence from other environmental parameters. These results from the modern Arctic help inform interpretations of past, terrestrially-derived plant waxes in Arctic depositional settings by reaffirming these fundamental assumptions about plant wax production and stable isotope fractionation in paleoclimate research.

*Code and data availability.* The code and data used for analysis and producing figures for this study are available on Zenodo (Lindberg,
2025). A separate DOI for the data will be available soon once it is archived at the National Science Foundation Arctic Data Center.

*Author contributions.* KRL and EKT designed the study. MKR, HB, JHR, and KRL carried out the field sampling. EKT, MKR, and KRL funded the research. KRL conducted laboratory work for producing the data. KRL analyzed the data and provided interpretations of the results. KRL produced the Python code for data analysis and figure production. KRL wrote the manuscript with input from all co-authors.

*Competing interests.* The authors declare that they do not have any competing interests.



*Acknowledgements.* This research was made possible by funding from the National Science Foundation (NSF ARCSS Grant No. 1737716 and EAR-IF Grant No. 1652274 to EKT) and the Geological Society of America's graduate student research grant program. We thank Owen C. Cowling, Nancy Leon, Haben Berhe, Caleb K. Walcott-George, and Emily Earl for their technical support. We also thank the PACEMAP (Predicting Arctic Change through Ecosystem MoleculAr Proxies) team for their insight and support.

Land Use Acknowledgement: We thank the Inuit communities of Baffin Island for allowing the PACEMAP team to conduct research on
their land and for providing field work logistical support. We recognize that research at the University at Buffalo took place on unceded Haudenosaunee/Six Nations Confederacy land.



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
