# Peer review of "Ecological and environmental controls on plant wax production and stable isotope fractionation in modern terrestrial Arctic vegetation"

_EGUsphere, 2025_

## Author Comment (AC1)

**Response to Anonymous Referee #1**

We thank the reviewer for their detailed comments and suggested considerations to improve our manuscript. We have addressed each individual comment below and will edit the manuscript accordingly. Reviewer comments are in italics and our responses are in normal text.

**Summary:**

Lindberg et al. investigate patterns in leaf wax (n-alkane and n-alkanoic acid) distribution in vegetation from the Arctic. The work provides a foundation for interpreting paleoclimate records based on leaf wax molecular proxies in sediment cores. Their assessment of the environmental parameters (meteorological/environmental effects, and vital effects of different species) is thorough, in that it combines their large dataset form the Eastern Canadian Arctic with a pan-Arctic synthesis. Ultimately, they provide a practical tool for paleoclimatologists working in the Arctic.

The manuscript is very clearly written, includes a solid statistically-based discussion, includes pertinent citations, and is essentially ready for publication, although I have two considerations for the authors, as well as a few very minor comments.

**Considerations:**

The first consideration is that latitude is not tested as an environmental parameter. We don't typically think of latitude as a driving factor, but in the case of Arctic leaf waxes, there has been discussion on if day-length impacts leaf wax hydrogen isotopes. Perhaps their study sites don't span a substantial gradient in day-length, or the length of the 24-hour daylight season (latitude is a proxy for this), but I think it would be worth at least acknowledging this point. That is, a previous study from Baffin Island by Shanahan et al. (2013) had anomalously small fractionation values between precipitation and leaf wax D/H – how does that previous. In fact, this is one paper that seems like it should be cited, or explained why it is not included in the synthesis.

We agree with the reviewer that latitude should be included in our statistical analyses of environmental parameters and plant wax indices. We will include text in the Materials and Methods, Results, and Discussion sections which incorporate the explanation of latitude as a proxy for the length of the 24-hour daylight season (Shanahan et al., 2013; Yang et al., 2009, 2011) and how this mechanism compares to correlations between latitude and plant wax indices. We will also adjust Figures 6-9 to include the results of these correlations.

In response to the second part of this consideration, we do not include plant waxes from Shanahan et al. (2013) in our pan-Arctic data synthesis because their data was extracted from lake sediments, not modern plants. We will adjust the wording as follows in our description of the data compilation methods in this study (Section 2.1) to better emphasize that we only compiled plant wax data extracted from terrestrial plants:

"To expand the sample size and range of environmental conditions, we compiled published plant wax data from terrestrial plants from sampling sites across the entire Arctic; within the latitude range spanned by the ECA transect (Fig. 1a)."

The second consideration is that among all the environmental parameters tested, they did not include Vapor Pressure Deficit. I wonder, if they combine the air temperature, relative humidity, and added in the modeled leaf temperature, would they find strong gradients in VPD across their sites, and how would this relate to the epsilon value. It has been untested, to my knowledge, but could be potentially revealing as an important environmental control.

We thank the reviewer for their suggestion to consider Vapor Pressure Deficit (VPD) in our analyses of Arctic environmental parameters. However, after testing VPD in our correlation matrices, we have decided not to include it in our revisions because VPD was strongly negatively correlated ( $r \le -0.92$ ) with relative humidity in all plant wax data subsets shown in Figures 6-9. This resulted in all Pearson r-values between VPD and plant wax indices having nearly the same magnitude with the opposite sign compared to correlations with relative humidity, so showing tests with one parameter reliably predicts the results of the other. Below, we have shown the derivation of VPD and its inverse relationship with relative humidity based on Equation 1 from our manuscript (Alduchov and Eskridge, 1996), which calculates relative humidity based on temperature (T) and dew point temperature (Dp):

Saturation Vapor Pressure: Es =  $e^{[(17.625 * T)/(243.04 + T)]}$  (Alduchov and Eskridge, 1996)

Actual Vapor Pressure: Ea =  $e^{[(17.625 * Dp)/(243.04 + Dp)]}$  (Alduchov and Eskridge, 1996)

Relative Humidity = (Ea/Es) \* 100

Vapor Pressure Deficit: VPD = Es - Ea

VPD = Es - (RH \* Es/100)

Additionally, incorporating modeled leaf temperature into our statistical analyses is beyond the scope of this study. This process would require implementing a separate model which would need to be run for each plant sample. Commonly used leaf models, such as Tealeaves (Muir, 2019) and NicheMapR (Kearney and Porter, 2020), are also not currently suited for modeling non-vascular plants, which represent a significant portion of our total samples.

**Minor Comments:**

Paragraph at line 244, which refers to Figure S2: Specify again in this paragraph that this includes data points from all plant types. Also, you mention that the pan-Arctic dataset has an n=386. But it does not look like Figure S2 has 386 data points. Can you clarify what is included in this figure?

We will clarify in this sentence and in the Figure S2 caption as follows that these Pearson correlations were performed using data from all plant growth forms in the pan-Arctic dataset. We also agree that our description of sample sizes in this manuscript needs clarification.

While we used data from a total of 386 plant samples between the samples we analyzed and compiled from other publications, not every data type was available for each sample or compound class. To illustrate this better, we will add two new tables to the supplement (to be Table S1 and Table S2) that show the number of data points for each data type in each compound class. Table S1 shows the raw sample counts and Table S2 shows the sample counts where plants of the same species collected at the same time from the same site are averaged together (see Section 2.4):

Section 3.3: "In the pan-Arctic dataset using all plant growth forms, we found that n-alkanoic acid  $\delta^2 H$  had a weak, positive correlation (r = 0.25; p = 0.02) with MAF precipitation  $\delta^2 H$  (Fig. S2a), while n-alkane  $\delta^2 H$  had a moderate, positive correlation (r = 0.64; p < 0.01) with MAF precipitation  $\delta^2 H$  (Fig. S2b)."

Figure S2 caption: "Pearson correlations and linear regressions in each panel were performed on the pan-Arctic dataset using all plant growth forms (see Table S2 for sample sizes)."

Figure 7 caption: specify if this is the pan-Arctic dataset or the ECA dataset. (It's stated in the text, but would help clarify the figure caption.)

All of the correlation matrices in Figures 6-9 were produced using the pan-Arctic dataset. We will add this clarification to each of those figure's captions.

**References:**

Alduchov, O. A. and Eskridge, R. E.: Improved Magnus Form Approximation of Saturation Vapor Pressure, J. Appl. Meteorol., 35, 601–609, https://doi.org/10.1175/1520-0450(1996)035%253C0601:IMFAOS%253E2.0.CO;2, 1996.

Kearney, M. R. and Porter, W. P.: NicheMapR – an R package for biophysical modelling: the ectotherm and Dynamic Energy Budget models, Ecography, 43, 85–96, https://doi.org/10.1111/ecog.04680, 2020.

Muir, C. D.: tealeaves: an R package for modelling leaf temperature using energy budgets, AoB PLANTS, 11, plz054, https://doi.org/10.1093/aobpla/plz054, 2019.

Shanahan, T. M., Hughen, K. A., Ampel, L., Sauer, P. E., and Fornace, K.: Environmental controls on the 2H/1H values of terrestrial leaf waxes in the eastern Canadian Arctic, Geochim. Cosmochim. Acta, 119, 286–301, https://doi.org/10.1016/j.gca.2013.05.032, 2013.

Yang, H., Pagani, M., Briggs, D. E. G., Equiza, M. A., Jagels, R., Leng, Q., and LePage, B. A.: Carbon and hydrogen isotope fractionation under continuous light: implications for paleoenvironmental interpretations of the High Arctic during Paleogene warming, Oecologia, 160, 461–470, https://doi.org/10.1007/s00442-009-1321-1, 2009.

Yang, H., Liu, W., Leng, Q., Hren, M. T., and Pagani, M.: Variation in n-alkane  $\delta D$  values from terrestrial plants at high latitude: Implications for paleoclimate reconstruction, Org. Geochem., 42, 283–288, https://doi.org/10.1016/j.orggeochem.2011.01.006, 2011.

---

## Author Comment (AC2)

**Response to Anonymous Referee #2**

We thank the reviewer for their thorough review of our manuscript and for providing helpful comments for improving our manuscript. We have addressed each individual comment below and will edit the manuscript accordingly. Reviewer comments are in italics and our responses are in normal text.

**Summary and Major Questions:**

Review of egusphere-2025-3849.

Ecological and environmental controls on plant wax production and stable isotope fractionation in modern terrestrial Arctic vegetation. By Kurt R. Lindberg, Elizabeth K. Thomas, Martha K. Raynolds, Helga Bültmann, and Jonathan H. Raberg

Dear associate editor and authors, I have read this manuscript with great pleasure, it is an interesting topic. I do have some questions though, some more scientific and some more technical. I will start with a scientific question, what about meltwater? If I understand the authors correctly they looked at relatively early growth, so when the snow and ice is still melting? The hydrogen isotope composition of snow is, as far as I know, different from rain. Plus, snow and ice maybe derived from other seasons than the precipitation during the growing season. Of course, different plant types might have different access to meltwater, some have roots and can access ground water that I guess in part is meltwater, some might live on rocks or trees and have less access to meltwater and some might live in bogs (or even "lakes" and have access to meltwater much longer time span. Could this affect the hydrogen isotopic composition of the different compounds measured and for instance have kept the 2H values much more stable than expected based on precipitation values?

The reviewer makes a very good point about the potential influence of meltwater on the  $\delta^2 H$  composition of soil water and, therefore, the  $\delta^2 H$  of terrestrial plant waxes. We will add text as follows to the Materials and Methods section (Section 2.2) about Arctic environmental parameters to explain our use of MAF precipitation to represent the  $\delta^2 H$  value of water that plants are using when synthesizing their plant waxes:

"We justify our use of MAF precipitation  $\delta^2H$  based on several studies which show that shallow Arctic soil water, from which most plants obtain their water, generally reflects growing season precipitation without the influence of  $^2H$ -depleted snowmelt (Chiasson-Poirier et al., 2020; Daniels et al., 2017; O'Connor et al., 2020; Sullivan and Welker, 2007). This is attributed to the soil still being frozen and impermeable during the snowmelt period, which causes the meltwater to be lost from the system as surface runoff (Woo, 2012)."

That brings me to my second comment or question. I really like the Bowen model and tool to reconstruct or estimate precipitation 2H values. However, the quality depends largely on the proximity of measuring stations. The interpolations between measuring stations can be very good if the landscape is fairly boring, as soon as you get elevation differences, for instance,

they might be off. This made me think that the strong dependence on epsilon values might over simplify things. Has this been considered?

We agree with the referee that using precipitation  $\delta^2H$  values from the OIPC comes with uncertainties related to a given site's surrounding topography and proximity to precipitation isotope measuring stations (e.g., Feakins and Sessions, 2010; Gorbey et al., 2022). We reference these some of these uncertainties in Sections 2.2 and 4.4 when justifying our use of OIPC precipitation isotopes and ERA5 reanalysis data for other environmental parameters since our ECA study sites, along with many others in the pan-Arctic dataset, do not have such data collected in-situ or from a close enough monitoring station. We will add additional text to these sections to further clarify why we chose to use OIPC precipitation isotope data and that we acknowledge its associated uncertainties in exchange for providing a consistent methodology for obtaining this data.

The authors give an overview of things hat might affect the 2H values, the one I was missing was the amount effect. I don't know if that plays a role in the settings discussed here, but I think there are elevation differences, perhaps there are also areas with way more precipitation than others? By the way, also a reason why sometimes the 2H model might have the 2H of precip. wrong, the amount effect.

Many studies have shown that variations in Arctic precipitation isotope ratios are primarily driven by seasonal changes in temperature and moisture source, instead of the precipitation amount effect which is more prominent in lower latitude sites (Broadman et al., 2020; Cluett et al., 2021; Dansgaard, 1964; Putman et al., 2017). We will add text as follows to the Materials and Methods section (Section 2.2) about Arctic environmental parameters that states these main drivers of Arctic precipitation isotope variability:

"Additionally, variations in Arctic precipitation isotope ratios are primarily driven by consistent, seasonal changes in temperature and moisture source, rather than amount-driven fractionation during individual precipitation events (Broadman et al., 2020; Cluett et al., 2021; Dansgaard, 1964; Putman et al., 2017)."

My last main comment is on the M and M section, I think the technical section the 2H measurements is a bit light. What reference materials were measured on the same machine to ensure data quality? Did the authors consider that peaks have to have a decent size for a reliable measurement? Measuring samples or compounds multiple times at roughly the same low peak height (or area) will give you reproducible values, not necessarily the "correct" values. I agree that this doesn't need to be in the manuscript, but the lack of technical details made me wonder. Of course, the authors have put a lot of effort in the statistical analysis of the results, but if the quality of the results can not be judged by the reviewer and other readers the statistics are also not so useful.

We will add additional text to the Materials and Methods section (Section 2.3) as follows on our analysis of plant wax n-alkanoic acid  $\delta^2H$  and  $\delta^{13}C$ . All GC-IRMS sequences were run with standards of known isotopic values to correct for instrument drift and peak size linearity. These methods are consistent with, and also described in, other studies where plant wax n-alkanoic acid stable isotope data was produced in the University at Buffalo Organic and

Stable Isotope Biogeochemistry Laboratory (Gorbey et al., 2021; Hollister et al., 2022; Holtzman et al., 2025):

"Stable isotope analysis was conducted in the University at Buffalo Organic and Stable Isotope Biogeochemistry Laboratory using a Thermo Delta V+ Isotope Ratio Mass Spectrometer (IRMS) with a split/splitless injector and a TriPlus RSH autosampler, connected to the IRMS via IsoLink II and Conflo IV with all samples and standards run in triplicate. Within each IRMS sequence, we ran standards of  $C_{20}$  and  $C_{28}$  n-alkanoic acids to calibrate sample  $\delta^2$ H results to the Vienna Standard Mean Ocean Water (VSMOW) scale and to correct for chromatograph peak size linearity. We also used standards of  $C_{18}$  and  $C_{24}$  n-alkanoic acids to correct for instrument drift."

In an Australian study, I think leaf water enrichment and relative humidity were determined to be the most important, I think. They measured along a transect with very little difference in source water 2H and came to that conclusion. I think a paper by Ansgar Kahmen and or his group. Just a suggestion, I know very different environment.

We thank the reviewer for bringing our attention to this study. We will reference this study in our discussion in Section 4.4 on the expected vs. observed relationship between terrestrial Arctic plant wax  $\epsilon_{app}$  and relative humidity.

Overall, I think the manuscript needs some work, perhaps different water sources, (including precipitation, ice, snow, meltwater, lake water etc.) have been measured and can be compared to the model results for possibly more accurate epsilon values? I definitely would like to be able to better judge the quality of the 2H measurements, so a bit more detail I the M en M section would be appreciated. If these questions have been addressed I think the manuscript is very publishable.

**Some more detailed remarks:**

Line 4: the govern stable hydrogen isotope fractionation?

This statement about ecological and environmental controls on stable isotope fractionation apply to both hydrogen and carbon isotopes, even though this study focuses primarily on the fractionation of variable precipitation hydrogen isotopes. That said, we will clarify that, in the context of plant waxes, we are referring to the fractionation of "their" stable isotopes.

Line 16 and 17: independent of precipitation but reflecting source water 2H, so are precipitation and source water different things? Precipitation amount?

We will alter the wording in this sentence as follows to clarify that we are referring to precipitation  $\delta^2 H$  in the context of terrestrially-derived Arctic plant waxes:

"Instead, changes in terrestrially-derived sedimentary plant wax distributions reflect changes in plant taxa present through time, and changes in terrestrially-derived plant wax  $\delta^2$ H reflect change in precipitation  $\delta^2$ H."

Line 35: Baas et al. A comparative study of lipids in Sphagnum species (2000) Organic Geochemistry 31 535- 541. Ficken et al. An n-alkane proxy for the sedimentary input of submerged/ floating freshwater aquatic macrophytes (2000) Organic Geochemistry 31 745-749.

We will add the citations listed in this comment (Baas et al., 2000; Ficken et al., 2000) to the end of this sentence.

Line 45: higher and lower d13C values rather than more or less depleted or enriched?

We prefer to use the terms "enriched" and "depleted" when comparing stable isotope values to avoid confusion with their (usually) negative measurement values. To account for both frames of reference in this sentence, we will add explanations of these terms relative to "higher" and "lower"  $\delta^{13}$ C values as follows:

"The difference in carbon isotope fractionation between more  $^{13}$ C-depleted (lower  $\delta^{13}$ C values)  $C_3$  and more  $^{13}$ C-enriched (higher  $\delta^{13}$ C values)  $C_4$  photosynthetic pathways has been well documented, and is commonly used to reconstruct past vegetation change between these two broad plant community types (Cerling and Harris, 1999), although  $C_4$  plants do not occur at high latitudes."

Line 57: the precipitation amount, the amount effect?

In this sentence, we are just referring to the "total" precipitation amount that a given sampling site receives (annual, seasonal, etc.), not the amount effect related to precipitation stable isotopes. We will modify the wording in this sentence as follows to clarify this point:

"Environmental factors, including temperature, total precipitation amount, and relative humidity, may also affect how individual plant taxa produce different plant wax chain-lengths and fractionate stable carbon and hydrogen isotopes."

Figure legend of figure 2, precipitation is in 2e not 2d.

We will correct the reference to panel e in the last sentence of this figure caption.

Line 164: I assume the H3+ factor slowly changed from the one to the other value over a significant amount of time and this was not the day to day variation? I do like that you mention the H3+ factor, lots of people have stopped doing that.

We will add text to this section (Section 2.3) as follows to clarify that ECA plant wax n-alkanoic acid  $\delta^2 H$  measurements were taken in two batches, one during the Fall of 2022 and the other during the Summer of 2023. During both of these periods,  $H_3^+$  were very consistent between GC-IRMS sequences, varying by only  $\sim 0.1$ . The shift in  $H_3^+$  values between the two measurement periods was caused by instrument maintenance and switching between analyzing for  $\delta^2 H$  and  $\delta^{13} C$ :

"Sequences run in the Fall of 2022 had  $H_3^+$  factors ranging from 4.873  $\pm$  0.025 to 5.005  $\pm$  0.038 (mean  $\pm$  1 $\sigma$ ) and  $H_3^+$  factors in sequences run during the Summer of 2023 ranged from 3.400  $\pm$  0.055 to 3.509  $\pm$  0.016."

Line 247: an average of 30.5 plus or minus 18.6 permil. What drives this range?

We thank the reviewer posing this very good question about what drives these large ranges in  $\delta^2 H$  between plant wax chain-lengths from the same sample. Other studies have noted the variability in plant wax  $\delta^2 H$  and  $\delta^{13} C$  between chain-lengths and have attributed it to systemic changes in isotope fractionation during the production of different chain-lengths (Chikaraishi and Naraoka, 2007; Feakins et al., 2016; Saishree et al., 2023), although the exact mechanisms are not well understood. Like those studies, this variability is present in all data produced and compiled for this study, without a clear pattern between chain-length and degree of isotope enrichment or depletion. The mean and standard deviation of stable isotope ranges from our ECA n-alkanoic acid samples are greater than the pan-Arctic dataset, which may be due to our inclusion of more non-vascular plants (mosses, liverworts, lichens) which were sparsely sampled from other Arctic sites. We will add text as follows to the Results and Discussion sections (Section 3.2, 3.3, and 4.4) comparing this variability between our new ECA data and the pan-Arctic dataset along with the potential mechanism described above:

Section 3.2: "The range in ECA n-alkanoic acid  $\delta^{13}$ C between chain-lengths within an individual sample was an average of 3.2  $\pm$  2.3‰, which was greater than the average ranges in pan-Arctic n-alkanoic acids (2.6  $\pm$  1.9‰) and n-alkanes (1.3  $\pm$  0.6‰)."

Section 3.3: "The individual sample range in ECA n-alkanoic acid  $\delta^2 H$  between chain-lengths was an average of 31.9  $\pm$  18.3%, which was greater than the average ranges in pan-Arctic n-alkanoic acids (25.3  $\pm$  14.3%) and n-alkanes (17.0  $\pm$  13.4%)."

Section 4.4: "Saishree et al. (2023) and other studies (Chikaraishi and Naraoka, 2007; Feakins et al., 2016), also note the substantial variability we observe in plant wax  $\delta^2H$  and  $\delta^{13}C$  between chain-lengths from the same sample and have attributed it to systemic changes in biosynthetic fractionation as plants produce different chain-length waxes."

Line 294: this is from the Bowen model, right, not measured? I know measurements also have their issues.

Yes, these precipitation  $\delta^2H$  ranges are from the Online Isotopes in Precipitation Calculator (Bowen et al., 2005; Bowen and Revenaugh, 2003). We will add that clarification as follows in this sentence. We also realized that this information was only initially introduced in the Figure

2 caption, and will add similar text to the end of the Materials and Methods section (Section 2.2) on Environmental Parameters and Precipitation Isotopes.

Section 3.5: "Temperatures ranged from 3.6 to 12.4 °C (Fig. 2a), total precipitation ranged from 130.3 to 803.5 mm (Fig. 2b), relative humidity ranged from 56.3 to 80.1% (Fig. 2c), elevation ranged from 0 to 950 masl (Fig. 2d), and OIPC-derived precipitation  $\delta^2$ H ranged from -145.5 to -86.6% (Fig. 2e).

Section 2.2: "We calculated MAF average amount-weighted precipitation isotope  $\delta^2 H$  using monthly average  $\delta^2 H$  from the Online Isotopes in Precipitation Calculator (Bowen et al., 2005; Bowen and Revenaugh, 2003) and ERA5 precipitation amount."

Line 378-379: So, what about meltwater?

As stated in our response to the previous comment about meltwater, we will add text to Section 2.2 justifying our assumption that Arctic terrestrial plants use soil water that reflects growing season precipitation. Will will also modify this sentence as follows to reiterate that we are referring to water availability during the growing season when plants are synthesizing their cuticular waxes:

"Cold summer temperatures and restricted soil drainage due to bedrock and permafrost minimize water limitations during the growing season in the Arctic (Gold and Bliss, 1995), the period in which plant wax synthesis occurs (Tipple et al., 2013)."

Line 441: enriched and depleted relative to what? Each other? I love isotope lingo, but there are a lot of potential readers out there that get confused with all these relative terms. It might be better to use delta values and lower and higher.

Similar to our response to the comment for Line 45, we prefer to use "enriched" and "depleted" terminology when comparing stable isotope values. However, we agree that the use of the word "relative" in this sentence is very ambiguous and will rephrase this sentence as follows to better explain our point about the  $\delta^{13}$ C values of shrubs:

"Shrubs had the most significantly different distributions of n-alkanoic acid  $\delta^{13}$ C values based on Mann-Whitney U tests (Fig. 4e)."

Line 464-466: Not only the mean annual is important, also the amount that falls in one session (again the amount effect) is important. That is a measure that is not always captured by yearly means and averages, every day a little bit or everything in just 2 days, it makes a difference for the plants, sure, but also for the 2H of the precipitation. Apparent fractionation that varies with 600 mm?

While the reviewer makes a great point about the potential influence of the distribution of precipitation throughout a given season, our ERA5 reanalysis datasets (Hersbach et al., 2020) are limited to monthly precipitation amounts from each year, preventing us from investigating higher-resolution precipitation variability. However, as we previously stated in our

responses, the amount effect does not exert a strong influence on Arctic precipitation  $\delta^2 H$  compared to variability in moisture source and temperature which are more seasonally consistent (Broadman et al., 2020; Cluett et al., 2021; Dansgaard, 1964; Putman et al., 2017). To address the second part of this comment, we will modify the wording in this sentence as follows to clarify that the referenced correlation between apparent fractionation and mean annual precipitation amount had a lot of scatter in the data points between 30 and ~600 mm compared to the entire range of mean annual precipitation values in Liu et al. (2023):

"However, mean annual precipitation amounts in their sampling locations ranged from 30 to 1720 mm, with much more scatter in  $\epsilon_{app}$  values in the range of 30 to ~600 mm of mean annual precipitation (Liu et al., 2023)."

Line 469: Could that be, to some degree, an effect of using the model and it not capturing al the variability there actually is I precipitation 2H? In figure 2d I noticed quite some elevation differences, for instance.

Yes, it is possible that some of the actual variability in precipitation  $\delta^2H$  is lost by using values from the OIPC instead of *in-situ* measurements due to local topography and/or regional orographic distillation of the site's moisture sources. However, as we stated in our response to the previous comment about OIPC uncertainties and in the manuscript, we do not have site-specific seasonal precipitation isotope data available to use instead. We will add this statement as follows to the end of this paragraph that specifically acknowledges this point about the potential loss in precipitation isotope variability.

"It is possible that the ranges of MAF precipitation amount, temperature, and relative humidity in the pan-Arctic dataset are not enough to drive consistent, significant changes in  $\epsilon_{app}$  (Feakins and Sessions, 2010). Though, it is also possible that some of the total  $\epsilon_{app}$  variability is lost by using spatially interpolated OIPC precipitation isotope  $\delta^2$ H instead of *in-situ* measurements (Feakins and Sessions, 2010; Gorbey et al., 2022)."

**References:**

Baas, M., Pancost, R., Van Geel, B., and Sinninghe Damsté, J. S.: A comparative study of lipids in Sphagnum species, Org. Geochem., 31, 535–541, https://doi.org/10.1016/S0146-6380(00)00037-1, 2000.

Bowen, G. J. and Revenaugh, J.: Interpolating the isotopic composition of modern meteoric precipitation, Water Resour. Res., 39, 2003WR002086, https://doi.org/10.1029/2003WR002086, 2003.

Bowen, G. J., Wassenaar, L. I., and Hobson, K. A.: Global application of stable hydrogen and oxygen isotopes to wildlife forensics, Oecologia, 143, 337–348, https://doi.org/10.1007/s00442-004-1813-y, 2005.

Broadman, E., Kaufman, D. S., Henderson, A. C. G., Malmierca-Vallet, I., Leng, M. J., and Lacey, J. H.: Coupled impacts of sea ice variability and North Pacific atmospheric circulation

- on Holocene hydroclimate in Arctic Alaska, Proc. Natl. Acad. Sci., 117, 33034–33042, https://doi.org/10.1073/pnas.2016544117, 2020.
- Cerling, T. E. and Harris, J. M.: Carbon isotope fractionation between diet and bioapatite in ungulate mammals and implications for ecological and paleoecological studies, Oecologia, 120, 347–363, https://doi.org/10.1007/s004420050868, 1999.
- Chiasson-Poirier, G., Franssen, J., Lafrenière, M. J., Fortier, D., and Lamoureux, S. F.: Seasonal evolution of active layer thaw depth and hillslope-stream connectivity in a permafrost watershed, Water Resour. Res., 56, e2019WR025828, https://doi.org/10.1029/2019WR025828, 2020.
- Chikaraishi, Y. and Naraoka, H.:  $\delta 13C$  and  $\delta D$  relationships among three n-alkyl compound classes (n-alkanoic acid, n-alkane and n-alkanol) of terrestrial higher plants, Org. Geochem., 38, 198–215, https://doi.org/10.1016/j.orggeochem.2006.10.003, 2007.
- Cluett, A. A., Thomas, E. K., Evans, S. M., and Keys, P. W.: Seasonal Variations in Moisture Origin Explain Spatial Contrast in Precipitation Isotope Seasonality on Coastal Western Greenland, J. Geophys. Res. Atmospheres, 126, e2020JD033543, https://doi.org/10.1029/2020JD033543, 2021.
- Daniels, W. C., Russell, J. M., Giblin, A. E., Welker, J. M., Klein, E. S., and Huang, Y.: Hydrogen isotope fractionation in leaf waxes in the Alaskan Arctic tundra, Geochim. Cosmochim. Acta, 213, 216–236, https://doi.org/10.1016/j.gca.2017.06.028, 2017.
- Dansgaard, W.: Stable isotopes in precipitation, Tellus Dyn. Meteorol. Oceanogr., 16, 436, https://doi.org/10.3402/tellusa.v16i4.8993, 1964.
- Feakins, S. J. and Sessions, A. L.: Controls on the D/H ratios of plant leaf waxes in an arid ecosystem, Geochim. Cosmochim. Acta, 74, 2128–2141, https://doi.org/10.1016/j.gca.2010.01.016, 2010.
- Feakins, S. J., Bentley, L. P., Salinas, N., Shenkin, A., Blonder, B., Goldsmith, G. R., Ponton, C., Arvin, L. J., Wu, M. S., Peters, T., West, A. J., Martin, R. E., Enquist, B. J., Asner, G. P., and Malhi, Y.: Plant leaf wax biomarkers capture gradients in hydrogen isotopes of precipitation from the Andes and Amazon, Geochim. Cosmochim. Acta, 182, 155–172, https://doi.org/10.1016/j.gca.2016.03.018, 2016.
- Ficken, K. J., Li, B., Swain, D. L., and Eglinton, G.: An n-alkane proxy for the sedimentary input of submerged/floating freshwater aquatic macrophytes, Org. Geochem., 31, 745–749, https://doi.org/10.1016/S0146-6380(00)00081-4, 2000.
- Gold, W. G. and Bliss, L. C.: Water Limitations and Plant Community Development in a Polar Desert, Ecology, 76, 1558–1568, https://doi.org/10.2307/1938157, 1995.
- Gorbey, D. B., Thomas, E. K., Crump, S. E., Hollister, K. V., Raynolds, M. K., Raberg, J. H., de Wet, G., Sepúlveda, J., and Miller, G. H.: Southern Baffin Island mean annual precipitation isotopes modulated by sumemr and autumn moisture source changes during the past 5800 years, J. Quat. Sci., 37, 967–978, https://doi.org/10.1002/jqs.3390, 2021.

- Gorbey, D. B., Thomas, E. K., Sauer, P. E., Raynolds, M. K., Miller, G. H., Corcoran, M. C., Cowling, O. C., Crump, S. E., Lovell, K., and Raberg, J. H.: Modern Eastern Canadian Arctic Lake Water Isotopes Exhibit Latitudinal Patterns in Inflow Seasonality and Minimal Evaporative Enrichment, Paleoceanogr. Paleoclimatology, 37, https://doi.org/10.1029/2021PA004384, 2022.
- Hersbach, H., Bell, B., Berrisford, P., Hirahara, S., Horányi, A., Muñoz-Sabater, J., Nicolas, J., Peubey, C., Radu, R., Schepers, D., Simmons, A., Soci, C., Abdalla, S., Abellan, X., Balsamo, G., Bechtold, P., Biavati, G., Bidlot, J., Bonavita, M., De Chiara, G., Dahlgren, P., Dee, D., Diamantakis, M., Dragani, R., Flemming, J., Forbes, R., Fuentes, M., Geer, A., Haimberger, L., Healy, S., Hogan, R. J., Hólm, E., Janisková, M., Keeley, S., Laloyaux, P., Lopez, P., Lupu, C., Radnoti, G., De Rosnay, P., Rozum, I., Vamborg, F., Villaume, S., and Thépaut, J.: The ERA5 global reanalysis, Q. J. R. Meteorol. Soc., 146, 1999–2049, https://doi.org/10.1002/qj.3803, 2020.
- Hollister, K. V., Thomas, E. K., Raynolds, M. K., Bültmann, H., Raberg, J. H., Miller, G. H., and Sepúlveda, J.: Aquatic and Terrestrial Plant Contributions to Sedimentary Plant Waxes in a Modern Arctic Lake Setting, J. Geophys. Res. Biogeosciences, 127, e2022JG006903, https://doi.org/10.1029/2022JG006903, 2022.
- Holtzman, H., Thomas, E. K., Erb, M., Marshall, L., Castañeda, I. S., Kaufman, D., McKay, N. P., and Melles, M.: Early Holocene Atmospheric Circulation Changes Over Northern Europe Based on Isotopic and Biomarker Evidence From Kola Peninsula, Paleoceanogr. Paleoclimatology, 40, e2024PA005076, https://doi.org/10.1029/2024PA005076, 2025.
- Liu, H., Wang, S., Wang, H., Cao, Y., Hu, J., and Liu, W.: Apparent fractionation of hydrogen isotope from precipitation to leaf wax n-alkanes from natural environments and manipulation experiments, Sci. Total Environ., 877, 162970, https://doi.org/10.1016/j.scitotenv.2023.162970, 2023.
- O'Connor, K. F., Berke, M. A., and Ziolkowski, L. A.: Hydrogen isotope fractionation in modern plants along a boreal-tundra transect in Alaska, Org. Geochem., 147, 104064, https://doi.org/10.1016/j.orggeochem.2020.104064, 2020.
- Putman, A. L., Feng, X., Sonder, L. J., and Posmentier, E. S.: Annual variation in event-scale precipitation  $\delta 2$  H at Barrow, AK, reflects vapor source region, Atmospheric Chem. Phys., 17, 4627–4639, https://doi.org/10.5194/acp-17-4627-2017, 2017.
- Saishree, A., Managave, S., Sarangi, V., and Sanyal, P.: Experimental evidence suggests dominance of species effect on the variability in hydrogen isotope fractionation between leaf wax compounds and source water, Org. Geochem., 183, 104656, https://doi.org/10.1016/j.orggeochem.2023.104656, 2023.
- Shanahan, T. M., Hughen, K. A., Ampel, L., Sauer, P. E., and Fornace, K.: Environmental controls on the 2H/1H values of terrestrial leaf waxes in the eastern Canadian Arctic, Geochim. Cosmochim. Acta, 119, 286–301, https://doi.org/10.1016/j.gca.2013.05.032, 2013.

Sullivan, P. F. and Welker, J. M.: Variation in leaf physiology of Salix arctica within and across ecosystems in the High Arctic: test of a dual isotope ( $\Delta 13C$  and  $\Delta 18O$ ) conceptual model, Oecologia, 151, 372, https://doi.org/10.1007/s00442-006-0602-1, 2007.

Tipple, B. J., Berke, M. A., Doman, C. E., Khachaturyan, S., and Ehleringer, J. R.: Leaf-wax n -alkanes record the plant—water environment at leaf flush, Proc. Natl. Acad. Sci., 110, 2659—2664, https://doi.org/10.1073/pnas.1213875110, 2013.

Woo, M.: Permafrost Hydrology, Springer Berlin Heidelberg, Berlin, Heidelberg, https://doi.org/10.1007/978-3-642-23462-0, 2012.